# Separate lanes for adding and reading in the white matter highways of the human brain

Mareike Grotheer [1], Zonglei Zhen[2], Garikoitz Lerma-Usabiaga[1,3] & Kalanit Grill-Spector [1,4]

Math and reading involve distributed brain networks and have both shared (e.g. encoding of visual stimuli) and dissociated (e.g. quantity processing) cognitive components. Yet, to date, the shared vs. dissociated gray and white matter substrates of the math and reading networks are unknown. Here, we define these networks and evaluate the structural properties of their fascicles using functional MRI, diffusion MRI, and quantitative MRI. Our results reveal that there are distinct gray matter regions which are preferentially engaged in either math (adding) or reading, and that the superior longitudinal and arcuate fascicles are shared across the math and reading networks. Strikingly, within these fascicles, reading- and math-related tracts are segregated into parallel sub-bundles and show structural differences related to myelination. These findings open a new avenue of research that examines the contribution of sub-bundles within fascicles to specific behaviors.

[1] Psychology Department, Stanford University, Stanford, CA 94305, USA. [2] Beijing Key Laboratory of Applied Experimental Psychology, Faculty of Psychology, Beijing Normal University, Beijing, 100875, China. [3] BCBL. Basque Center on Cognition, Brain and Language, Mikeletegi Pasealekua 69, Donostia - San Sebastián 20009 Gipuzkoa, Spain. [4] Stanford Neurosciences Institute, Stanford University, Stanford, CA 94305, USA. Correspondence and requests for materials should be addressed to M.G. (email: grotheer@stanford.edu)

Math and reading are essential for functioning in modern society. While they are distinct tasks, math and reading utilize several overlapping cognitive processes, including encoding of visual stimuli, verbalization, and working memory[1]. There is also a high rate of co-morbidity between math and reading disabilities: up to 66% of children affected by dyscalculia, a math learning disability, also suffer from dyslexia, a reading learning disability[2]. These data suggest that math and reading may rely on shared neural substrates. The degree to which brain activations related to math and reading overlap may also be task-dependent. For example, responses related to arithmetic fact retrieval, e.g., during adding tasks involving small numbers, such as the one used in the current study, are proposed to overlap more extensively with responses related to reading than responses induced by procedural-based computations[3]. While a large body of research has examined both cortical regions and white matter connections of the reading network[4,5], the cortical regions and white matter connections of the math network and how they relate to the reading network are not well understood[6,7].

Research indicates that several white matter fascicles are key for reading. The fascicles of the reading network include: (i) The arcuate fasciculus (AF), which connects the frontal and temporal cortices. Diffusion MRI measurements show that fractional anisotropy (FA) in the left AF correlates with phonological awareness[8–10] and is reduced in dyslexics[9,11]. (ii) The inferior fronto-occipital fasciculus (IFOF), which connects the frontal and occipital cortices. Children with dyslexia show reduced leftward asymmetry of the IFOF[12] and FA of this tract is linked to orthographic processing skill[9]. (iii) The inferior longitudinal fasciculus (ILF), which connects occipital cortex with the anterior temporal cortex. Lesions to the ILF can lead to pure alexia[13], and atypical development of FA in the ILF is associated with poor reading proficiency[10,14]. (iv) The vertical occipital fasciculus (VOF), which connects the occipital and parietal cortices and is thought to relay top-down signals from the intraparietal sulcus (IPS) to ventral occipito-temporal cortex during reading[15]. Interestingly, these four fascicles also intersect with the visual word form area (VWFA)[16–18], a region in the occipito-temporal sulcus (OTS) that responds preferentially to words over other stimuli[19,20] and is causally involved in word recognition[21].

Presently, it is unclear if the fascicles associated with reading are also associated with math. This gap in knowledge is due to four reasons: First, substantially more research has been done on the neural bases of reading[4,5] than the neural bases of math[6]. Second, most prior studies have evaluated either the neural bases of math[7,22–24] or the neural bases reading[16,19,25–28], but not both systems within the same individuals (but see ref. [3]). Third, no study has examined the relation between white matter tracts and functional regions involved in math within the same participants (for related work see ref. [29]). Thus, the goal of this study is two-fold: (1) identify and quantify the white matter connections of cortical regions involved in math within typical adults, and (2) determine which aspects of this white matter are unique to the math network and which are shared with the reading network.

To accomplish these goals, we applied a multimodal approach in which we collected functional MRI (fMRI), diffusion MRI (dMRI), and quantitative MRI (qMRI) data in 20 typical adults. The fMRI experiment identified in each participant's brain the cortical regions that are involved in reading, adding, or both (Fig. 1b). As in our prior study[30], we presented participants with number-letter morphs, such that the three tasks, reading, adding or remembering colors, could be performed on identical stimuli (Fig. 1a). Next, using dMRI measurements and tractography with constrained spherical deconvolution[31], we determined the white matter tracts of the math and reading networks: First, we generated a white matter connectome for each participant. Then, using Automated Fiber Quantification (AFQ)[32], we identified 13 fascicles in each participant's brain, most of them in each hemisphere. Next, to identify functionally defined white matter tracts (fWMT) of the math and reading networks, we determined which white matter tracts either begin or terminate in the gray-white matter interface (GWMI) underneath functional regions of interest (fROIs) involved in reading, adding, or both (Fig. 1c). We also evaluated through which fascicles these fWMTs travel. Finally, using qMRI, we tested if there are differences in the white matter characteristics across the reading and math networks. QMRI[33,34] was used to measure proton relaxation time (T$_1$), which in the white matter correlates with myelination[35]. This enabled in vivo assessment of microstructural properties of the identified tracts.

Here we find that dissociated gray matter regions are involved in adding and reading. Further, while there are shared fascicles across the math and reading networks (arcuate and superior longitudinal fascicles), we find that distinct sub-bundles within these fascicles contribute to either math or reading. Thus, our data suggest that math and reading are processed largely in parallel in the adult brain.

## Results

**Neighboring gray matter regions process math and reading.** We used fMRI to define gray matter regions that are involved in math and reading in each participant. In the fMRI experiment, 20 adult participants performed a reading task, an adding task, and a color memory task on identical visual stimuli (number-letter morphs, Fig. 1a). In each trial, subjects viewed a cue indicating the task (Read/Add/Color), then viewed four number-letter morph stimuli that were presented sequentially, and at the end of the trial gave a 2-alternative-forced choice answer. In the adding task, subjects summed the stimuli and indicated which number corresponds to the correct sum; in the reading task, they read the stimuli and indicated which of the words had been presented; in the color task, they attended to the color of the stimuli and indicated which of the colors they had seen. Crucially, these tasks were matched in their working memory load and the amount of verbalization they elicit. Participants' performance in these tasks is summarized in the Methods.

We identified (i) regions involved in math, defined by higher responses during the adding task than the reading and color tasks (as in ref. [30]), (ii) regions involved in reading, defined by higher responses during the reading task than the adding and color tasks, and (iii) regions involved in both math and reading, defined by the conjunction of higher responses during adding than color task and higher responses during reading than color task. Crucially, all regions were defined in individual subjects' native anatomical space and without spatial smoothing, as both group averaging and spatial smoothing may introduce artificial overlap between regions[36].

We consistently found stronger responses during the reading task compared to adding and color tasks in four anatomical expanses (example subject: Fig. 1b-green; all subjects: Supplementary Fig. 6, 7): (i) A region in the occipito-temporal sulcus (OTS), which likely corresponds to the visual word form area (VWFA[19,20]), (ii) a region in the superior temporal sulcus (STS), which extended into the middle temporal gyrus, (iii) a region in the supramarginal gyrus (SMG), which we will refer to as SMGr as it is part of the reading network, and (iv) a region in the inferior frontal gyrus (IFG), which likely corresponds to Broca's Area. As activations during reading were lower and less frequent in the right than the left hemisphere, in the main manuscript we focus on the left hemisphere and right hemisphere data are

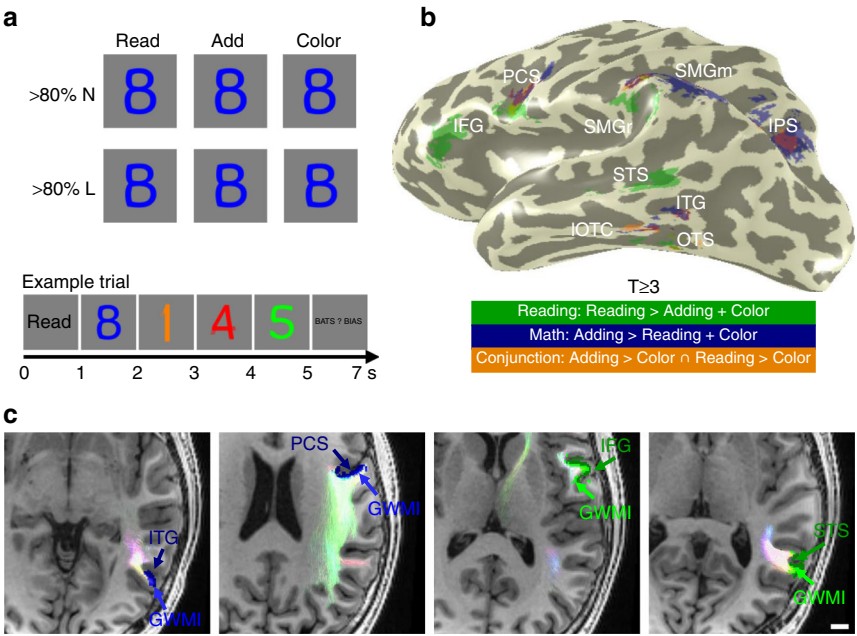

**Fig. 1** Identification of gray matter regions of the math and reading networks and their functionally-defined white matter tracts (fWMT). **a** FMRI experiment used to define math- and reading-related regions. Subjects viewed morphs between numbers and letters, containing either >80% letter (<20% number) or >80% number (<20% letter) information. At the beginning of each trial, a cue (Read/Add/Color) indicated which task should be performed, then four stimuli of the same morph type appeared for 1 s each, followed by an answer screen presented for 2 s. Subjects indicated their answer with a button press. Identical stimuli were presented across tasks. Trial structure is shown at the bottom. **b** Gray matter functional regions of interest (fROIs) of the math and reading networks. Green: Reading-related regions were defined based on higher responses in the reading task than other tasks; Blue: Math-related regions were defined based on higher responses in the adding task than other tasks; Orange: Regions that responded more strongly during reading vs. color and adding vs. color tasks. All fROIs were defined using a T ≥ 3 (voxel level) threshold in each participant's brain. **c** Example fROIs and their respective fWMTs in axial slices of a representative participant. Blue: Math fROIs. Green: Reading fROIs; lighter shades of blue and green under each fROI: respective gray-white-matter-interface (GWMI) of that fROI. The fiber tracts that terminate at the GWMI of each fROI are shown in pastel colors; the colors of the tracts indicate the main diffusion direction, pink: right/left; light green: anterior/posterior; light blue: superior/inferior. Scale bar in **c** indicates 1 cm. IFG inferior frontal gyrus, PCS precentral sulcus, SMGr reading fROI in supramarginal gyrus, SMGm math fROI in supramarginal gyrus, STS superior temporal sulcus, ITG inferior temporal gyrus, OTS occipito-temporal sulcus, IPS intraparietal sulcus, lOTC lateral occipito-temporal cortex

presented in the Supplementary Material (Supplementary Fig. 10, 12, 17, 21).

We also identified four bilateral regions that responded more strongly during the adding than reading and color tasks (example subject: Fig. 1b-blue; all subjects: Supplementary Fig. 6, 7): (i) a region in the inferior temporal gyrus (ITG), which is consistent with prior studies[30,37–39], (ii) a region in the intraparietal sulcus (IPS), consistent with research showing IPS involvement in numerosity processing[22,24], (iii) a region in the SMG, which we will refer to as SMGm, and (iv) a region in the precentral sulcus (PCS), near the inferior frontal junction, which has been implicated in visual object-based attention[40].

Interestingly, regions involved in reading and math often neighbored in the brain. In the prefrontal cortex, the reading-related IFG is proximal, but inferior to the math-related PCS. Likewise, in the SMG the reading-related SMGr is proximal, but inferior to the math-related SMGm. In the temporal cortex, math-related ITG is located between two reading-related fROIs, centered on the STS and OTS, respectively. In the IPS, we found only a math-related fROI.

Four regions in the brain showed higher responses during both the adding and reading tasks compared to the color task (conjunction fROIs: adding > color ∩ reading > color; Fig. 1b-orange, Supplementary Fig. 6, 7). These regions were located in (i) the IPS, (ii) the SMG, (iii) the PCS, and (iv) the lateral occipito-temporal cortex (lOTC), extending from the ITG to the OTS. Except for the lOTC region, conjunction fROIs were small and overlapped with the math fROIs. Indeed, responses in the

IPS, PCS, and SMG conjunction fROIs were significantly stronger during adding than reading (Supplementary Fig. 8; ANOVA with hemisphere, task, and stimulus as factors; main effect of task: IPS: $F(1,14) = 17.30$, $p = 0.001$, $\eta_p^2 = 0.55$; PCS: $F(1,16) = 12.97$, $p = 0.002$, $\eta_p^2 = 0.45$; SMG: $F(1,16) = 19.37$, $p = 0.0004$, $\eta_p^2 = 0.55$). The only conjunction fROI in which responses during adding and reading did not differ significantly was the lOTC (main effect of task: $F(1,17) = 3.83$, $p = 0.07$, $\eta_p^2 = 0.18$). This region overlaps with both a math fROI (in the ITG) and a reading fROI (in the OTS). Thus, in subsequent analyses we considered the IPS, PCS, and SMG regions as part of the math network, but the lOTC as a conjunction region involved in both tasks.

**The SLF and the AF contribute to math and reading networks**. After establishing which cortical regions are activated during adding and reading tasks, we determined which fascicles are associated with each of these fROIs (Fig. 2). Thus, using AFQ, we identified in each participant 13 well-established fascicles of the brain, most of them bilaterally (Supplementary Fig. 9). Next, we intersected each participant's classified white matter connectome with the GMWI underneath each of the fROIs to determine the functionally defined white matter tracts (fWMT) associated with reading and math (Fig. 1c). To summarize the fascicles connecting to each reading and math region across subjects, we quantified for each fROI what is the percentage of its fWMT that is associated with each of the 13 fascicles. We will refer to the relative contribution of each fascicle as connectivity weight.

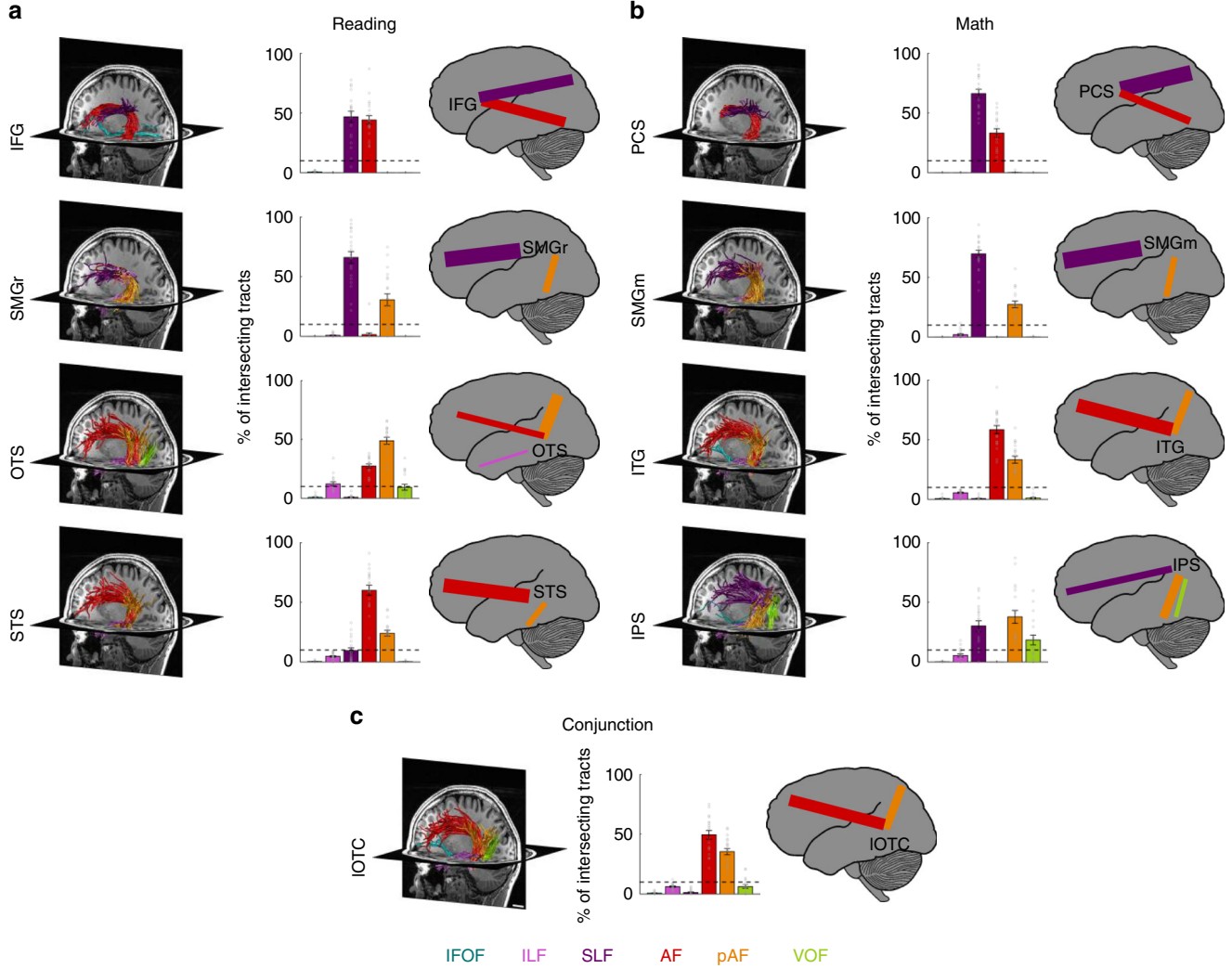

**Fig. 2** Functionally-defined white matter tracts (fWMT) of reading- and math-related regions. **a** Six fascicles (AF, SLF, pAF, VOF, ILF, and IFOF) contain >90% of all fWMT of the fROIs identified in the reading task. **b** The same six fascicles also contain >90% of all fWMT of the fROIs identified in the adding task. **c** lOTC conjunction fROI shows substantial connectivity with the AF and pAF. In **a**, **b**, **c**: Left: fWMT for each fROI in a representative subject's left hemisphere. The same subject is displayed in all panels; Fascicles are color coded in accordance with the legend at the bottom. Scale bar in **c** indicates 2 cm. Middle: Bar graphs showing what percentage of the fWMT is associated with each of the six fascicles. The graph shows the mean across subjects ± SEM. Circles: Individual subjects' data. Dashed horizontal line: Line is placed at 10%, which was the cutoff used for the schematics in the right columns. Right: Schematic illustration of the fascicles associated with each fROI. The thickness of the lines is derived from the bar graph, showing the relative connectivity weight of each fascicle. IFG inferior frontal gyrus, PCS precentral sulcus, SMGr reading fROI in supramarginal gyrus, SMGm math fROI in supramarginal gyrus, STS superior temporal sulcus, ITG inferior temporal gyrus, OTS occipito-temporal sulcus, IPS intraparietal sulcus, lOTC lateral occipito-temporal cortex, IFOF inferior fronto-occipital fasciculus, ILF inferior longitudinal fasciculus, SLF superior longitudinal fasciculus, AF arcuate fasciculus, pAF posterior arcuate fasciculus, VOF vertical occipital fasciculus

Results reveal two main findings. First, across participants, six fascicles contain almost all of the fWMT of math and reading fROIs (sum of their connectivity weights is above 90%). These fascicles are: IFOF, ILF, SLF, AF, posterior AF (pAF), and VOF. In Fig. 2, the left panels show the fWMT of math, reading, and lOTC conjunction fROIs in a representative subject, the middle panels show the connectivity weight of these fascicles across subjects, and the right panels provide a schematic illustration of the same data. Notably, three out of these six fascicles, the SLF, AF and pAF, form the backbone of the math and reading networks. For at least one fROI in each network, these fascicles contain >10% of all its fWMT. A similar pattern of results is observed in the right hemisphere (Supplementary Fig. 10), and when we controlled for fROI size, by repeating the analyses using constant-size spherical

ROIs (radius = 7 mm, centered on fROIs; Supplementary Fig. 11).

Second, we found that anatomically neighboring math and reading fROIs in the prefrontal cortex and SMG connect to the same fascicles with a comparable weight. That is, they show a relatively similar connectivity fingerprint. This is evident for the reading fROI in the IFG and the math fROI in the PCS that illustrate (i) substantial connectivity to the SLF (connectivity weight >46%), (ii) substantial connectivity to the AF (connectivity weight >33%), and (iii) no substantial connections to other fascicles (Fig. 2, first row). Similarly, both the reading and math fROIs in the SMG (SMGr and SMGm, respectively) show (i) strong connectivity to the SLF (weight >66%), (ii) connectivity to the pAF (>27%), and (iii) no substantial connections to other fascicles (Fig. 2, second row).

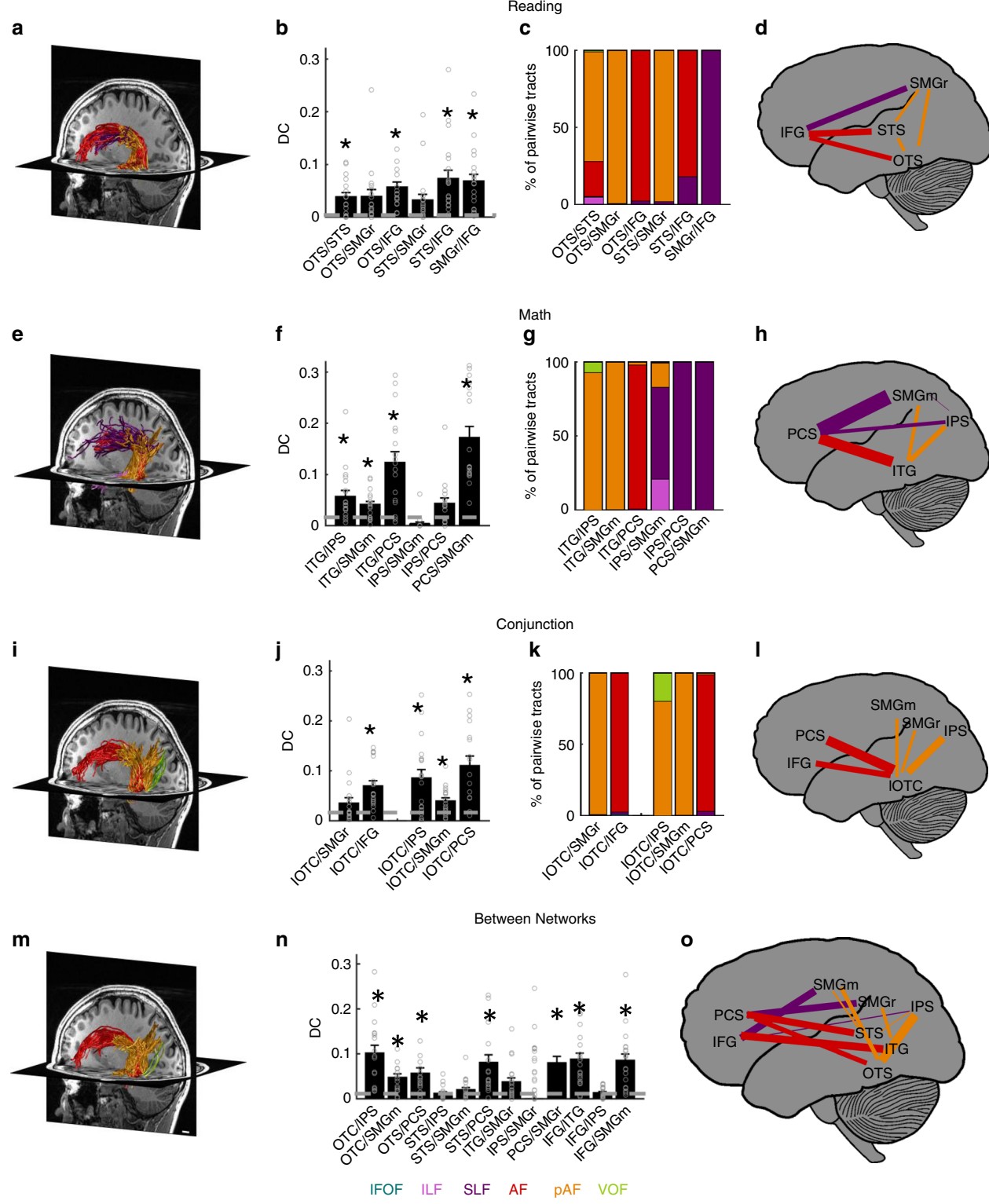

In comparison, reading and math fROIs in the temporal cortex and IPS showed more differentiated connections across networks. For example, while the reading fROI in the OTS showed above 10% connectivity weight with the AF, pAF, and ILF, a nearby math fROI in the ITG showed above 10% connectivity weight for the former two, but not the latter (Fig. 2, third row). Finally, the lOTC conjunction fROI was mainly connected to the AF (49.57%) and pAF (35.60%) (Fig. 2c).

Next, we evaluated within-network connections of the math and reading networks. To identify all pairwise connections, we intersected the fWMT of each fROI with the GWMI of each of the other non-neighboring fROIs within each network (Fig. 3a, e shows a representative subject). We quantified the pairwise connections relative to the total fWMT of each of the fROIs constituting the pair using the dice coefficient[41] (DC), and evaluated if it was significantly greater than chance level

**Fig. 3** Pairwise fWMT within and between the reading and math networks. **a-d** Within-network connections of the reading network. **e-h** Within-network connections of the math network. **i-l** Connections of the lOTC conjunction fROI to the math and reading networks. **m-o** Between-network connections. Left **a**, **e**, **i**, **m**: Pairwise white matter connections in a representative subject's left hemisphere. Scale bar in **m** indicates 1 cm. Second from left **b**, **f**, **j**, **n**: Dice coefficient (DC) of pairwise connections, mean across subjects ± SEM. The DC quantifies the overlap in the fWMT of both fROIs. X-labels indicate the fROI pairing. Dashed line: Chance level DC estimated from the average connections to out of network fROIs in ventral-temporal cortex that were activated maximally during the color task. Circles: Individual subjects' data. Asterisks: DC is significantly higher than chance (paired t-test, significance level was Bonferroni adjusted). Second from right in row 1-3 **c**, **g**, **k**: The relative contribution of six fascicles to the pairwise connections (legend at bottom). X-labels indicate the fROI pairing. Right **d**, **h**, **l**, **o**: Schematic illustration of the pairwise connections. Line thickness is scaled proportionally to the DC; Color indicates the fascicle with the highest relative contribution to the pairwise connection. IFG inferior frontal gyrus, PCS precentral sulcus, SMGr reading fROI in supramarginal gyrus, SMGm math fROI in supramarginal gyrus, STS superior temporal sulcus, ITG inferior temporal gyrus, OTS occipito-temporal sulcus, IPS intraparietal sulcus, lOTC lateral occipito-temporal cortex, IFOF inferior fronto-occipital fasciculus, ILF inferior longitudinal fasciculus, SLF superior longitudinal fasciculus, AF arcuate fasciculus, pAF posterior arcuate fasciculus, VOF vertical occipital fasciculus

(Fig. 3b, f). The $DC$ indicates the proportion of fWMT shared between two fROIs relative to the total fWMT of these fROIs.

In the reading network, we find significantly (Bonferroni-adjusted threshold of $p < 0.008$) above chance $DC$ between: (i) the OTS and the STS (paired $t$-test: $p = 0.0004$, t(17) = 4.37), (ii) the OTS and the IFG (paired $t$-test: $p < 0.0001$, t(17) = 5.51), (iii) the STS and the IFG (paired $t$-test: $p = 0.0003$, t(19) = 4.40), and (iv) SMGr and the IFG (paired $t$-test: $p < 0.0001$, t(19) = 5.05). The significant frontal-temporal connections of the reading network (STS-IFG; OTS-IFG) are supported by the AF, the frontal-parietal connections (IFG-SMGr) are supported by the SLF and the ventral-temporal connections (OTS-STS) are supported by the pAF (Fig. 3a-c).

In the math network, we find significantly (Bonferroni-adjusted threshold of $p < 0.008$) above chance $DC$ between (i) the ITG and the IPS (paired $t$-test: $p = 0.005$, t(18) = 3.22), which is supported by the pAF, (ii) the ITG and SMGm (paired $t$-test: $p = 0.001$, t(19) = 3.83), which is also supported by the pAF, between (iii) the ITG and the PCS (paired $t$-test: $p = 0.0001$, t(17) = 4.95), via the AF, and (iv) between SMGm and the PCS (paired $t$-test: $p < 0.0001$, t(17) = 7.02), through the SLF (Fig. 3e-g).

We summarize the pairwise connections and their predominant contributing fascicles in a schematic of within-network connections (Fig. 3d-reading, Fig. 3h-math). Overall, these analyses suggest that the math and the reading networks illustrate significant within-network connectivity, and the AF, pAF as well as the SLF emerge as key fascicles in both networks.

We also evaluated between-network connectivity in two ways: (i) by examining the pairwise connections of the conjunction lOTC fROI to each of the math and reading fROIs (Fig. 3i-l) and (ii) by evaluating the connections between pairs of fROIs across networks, where in each pair, one fROI was part of the reading network and the other fROI was part of the math network (Fig. 3m-o). Similar to the analyses of within-network connections, we only examined the long-range connections via fascicles, but not the local connections to neighboring fROIs.

For the lOTC conjunction fROI, we found significantly above chance $DC$ (Fig. 3i-l, Bonferroni-adjusted threshold of $p < 0.01$) only to one fROI of the reading network, namely the IFG (paired $t$-test: $p = 0.0001$, t(18) = 4.88, Fig. 3j), through the AF (Fig. 3k). In contrast, we found significantly above chance connections of the lOTC conjunction fROI to several fROIs of the math network: (i) the PCS (paired $t$-test: $p = 0.0002$, t(16) = 4.75), via the AF, (ii) SMGm (paired $t$-test: $p = 0.004$, t(18) = 3.29), via the pAF, and (iii) the IPS (paired $t$-test: $p = 0.0009$, t(18) = 3.93), also via the pAF (Fig. 3k). Connections to both networks are summarized in a schematic (Fig. 3l).

Analyzing pairwise connections across fROIs of the math and reading networks revealed significantly above chance $DC$ (Fig. 3n, Bonferroni-adjusted threshold of $p < 0.004$) between: (i) the OTS and the IPS (paired $t$-test: p < 0.0001, t(17) = 5.20), (ii) the OTS and SMGm (paired $t$-test: $p = 0.0004$, t(17) = 4.35), (iii) the

OTS and the PCS (paired $t$-test: $p = 0.002$, t(15) = 3.74), (iv) the STS and the PCS (paired $t$-test: $p = 0.0008$, t(17) = 4.06), (v) SMGr and the PCS (paired $t$-test: $p = 0.0002$, t(17) = 4.84), (vi) the IFG and the ITG (paired $t$-test: $p < 0.0001$, t(19) = 5.52), and (vii) the IFG and SMGm (paired $t$-test: $p < 0.0001$, t(19) = 5.18). Similar to the within-network connections described above, the pAF supported temporal-parietal between-network connections (OTS-IPS and OTS-SMGm), the SLF supported frontal-parietal between-network connections (SMGr-PCS and SMGm-IFG), and the AF supported frontal-temporal between-network connections (OTS-PCS, ITG-IFG). These connections are summarized in a schematic (Fig. 3o).

Analyses of the right hemisphere and with constant-size spherical ROIs (radius of 7 mm, centered on fROIs), show a similar pattern of results (Supplementary Fig. 12, 13).

**Reading and math tracts are segregated within SLF and AF.** The analyses in the prior section highlight the AF, pAF and the SLF as the main fascicles of both the math and the reading network. The pAF and AF also contribute to between-network connections. Nonetheless, within-network pairwise connectivity quantified by the $DC$ was significantly higher than between-network connectivity (paired $t$-test comparing within-network and between-network DCs: $p = 0.01$, t(15) = 2.83). Thus, in subsequent analyses, we will focus on within-network connections, and on connections of the lOTC conjunction fROI to both math and reading networks. We evaluated the SLF and the AF, as these emerged as the most prominent fascicles of both the math and the reading network.

Since the SLF and AF are large, it is unclear whether these entire fascicles are part of both networks or, alternatively, if sub-bundles within these fascicles relay tracts that support within-network connectivity of the reading and the math network, respectively. We tested these hypotheses by visualizing and quantifying within the SLF tracts connecting the IFG and SMGr in the reading network as well as the PCS and SMGm in the math network, and by examining if they are spatially intertwined or segregated in each subject. Similarly, in the AF, we visualized and compared tracts connecting the PCS and the ITG in the math network, with those connecting the IFG and STS in the reading network. Finally, within the AF, we also compared tracts that connect the lOTC conjunction fROI and the IFG (reading network) with those that connect the lOTC and the PCS (math network). Across subjects, and in both the SLF and the AF, our data showed that tracts were segregated by network: tracts of the math network (blue in Fig. 4a, d; all subjects: Supplementary Figs. 14-16) were consistently superior to tracts of the reading network (green in Fig. 4a, d; all subjects: Supplementary Figs. 14-16). Similarly, tracts connecting the lOTC to the PCS (math network) were consistently superior to tracts connecting the lOTC to the IFG (reading network) (Fig. 4g). This superior to

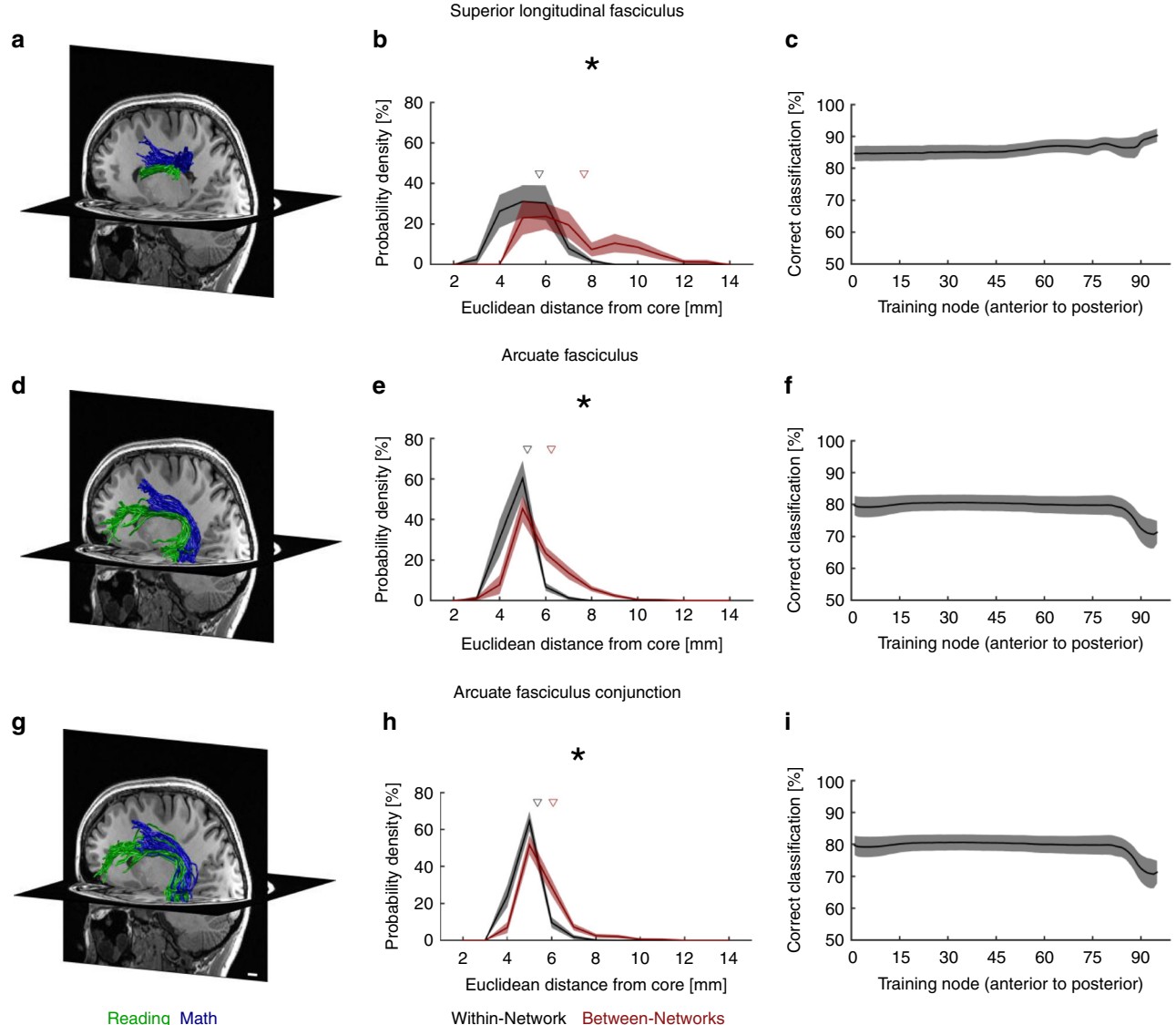

**Fig. 4** Pairwise connections of the reading and math networks are segregated and parallel in the SLF and the AF. **a–c**: SLF tracts connecting IFG and SMGr in the reading network and PCS and SMGm in the math network. **d–f**: AF tracts connecting the IFG and STS in the reading network and the PCS and ITG in the math network. **g**, **h**: AF tracts connecting the lOTC conjunction fROI with the IFG in the reading network and the PCS in the math network. **a**, **d**, **g**: Math (blue) and reading (green) tracts of the SLF and AF presented in the left hemisphere of a representative individual subject showing the spatial segregation of these tracts. Scale bar in **g** indicates 1 cm. **b**, **e**, **h**: Euclidean distance in millimeter (derived from x,y,z coordinates) of all tracts relative to the core (mean) tract, within-network (black) and between-networks (maroon). The distance was calculated across all tracts; the histograms show the distribution of distances across all nodes in each subject ± SEM. Upside-down triangles: Mean distance across nodes and subjects. Asterisk: Mean distances differ significantly, $p < 0.05$ (paired $t$-test). **c**, **f**, **i**: Performance of a linear SVM classifying math and reading tracts within the SLF and AF based on their spatial location. Data show mean classification accuracy across nodes ± SEM. IFG inferior frontal gyrus, PCS precentral sulcus, SMGr reading fROI in supramarginal gyrus, SMGm math fROI in supramarginal gyrus, STS superior temporal sulcus, ITG inferior temporal gyrus, lOTC lateral occipito-temporal cortex, AF arcuate fasciculus, SLF superior longitudinal fasciculus

inferior organization mirrors the spatial layout of neighboring math and reading fROIs on the cortical surface (Fig. 1b).

To validate and quantify this segregation, in each subject we sectioned the SLF and AF to 100 equal-sized bins, referred to as nodes, and conducted two additional analyses:

(1) We measured the distribution of distances (Euclidean distance in mm) of individual tracts from their within-network core (mean) tract vs. the core tract of the other network. We reasoned that if tracts of the math and reading networks are segregated, distances to the within-network core tract should be smaller than to the other network's core

tract. In contrast, if math and reading tracts are intertwined, these distances should not be significantly different. Results show that individual tracts are significantly closer to their within-network core tract compared to the other network's core tract in both the SLF (paired $t$-test on mean distance across tract: $p = 0.0006$, t(17) = 4.22) and the AF (paired $t$-test on mean distance across tract: $p < 0.0001$, t(17) = 6.12) (Fig. 4b, e). In the AF, tracts connecting the conjunction fROI in the lOTC with the IFG (reading network) and the PCS (math network) were also segregated (Fig. 4h, paired $t$-test on mean distance across tract: $p < 0.0001$, t(16) = 8.46).

(2)  We used an independent classifier approach to evaluate if reading- and math-related fWMTs are spatially segregated across the entire length of these fascicles or only in a restricted region. We reasoned that if they are segregated, a classifier should be able to determine if tracts belong to either the math or reading network based on their spatial location within the fascicle. To test this prediction, we trained a linear support vector machine (SVM) to distinguish math tracts from reading tracts based on their location (training: x,y,z coordinates of all tracts at a node). Then, we tested how well the SVM classifies independent data (at a fifth node posterior to each training node). Across subjects and nodes, classification of tracts as either being part of the reading or the math network was greater than 80% correct in the SLF and greater than 70% in the AF (Fig. 4c, f). In the AF, decoding accuracy dropped towards the posterior portion of the tract (Fig. 4f). Notably, the average classification was significantly higher than the 50% chance level (paired $t$-test: SLF: $p < 0.0001$, $t(17) = 17.47$; AF: $p < 0.0001$, $t(17) = 13.28$). When evaluating tracts within the AF that connect the lOTC conjunction fROI with the IFG (reading network) and the PCS (math network) (Fig. 4i), decoding accuracy again dropped towards the posterior end of the tract. Decoding accuracy still remained greater than 65%, and, on average, was significantly higher than chance level (paired $t$-test: $p < 0.0001$, $t(16) = 10.63$).

Similar results were also obtained for the right hemisphere (Supplementary Fig. 17), for constant-size spherical fROIs (Supplementary Fig. 18), and for direct fROI to fROI tractography (Supplementary Fig. 19). Overall, our analyses show that tracts associated with math and reading are segregated and run in parallel within the SLF and the AF, whereas tracts of the math network are located superior to tracts of the reading network.

**Reading tracts show faster $T_1$ than math tracts.** We next asked if there are structural differences between reading- and math-related tracts within the SLF and AF. To address this question, we used qMRI to determine $T_1$ of reading and math tracts. In the white matter, $T_1$ is inversely correlated with myelin content[35]. We also evaluated macromolecular tissue volume fraction (MTV), which indicates the fraction of non-water tissue in each voxel (Supplementary Fig. 20).

We first measured the average $T_1$ of math and reading tracts across the length of the fascicles. In the SLF, the average $T_1$ of reading-related tracts was significantly lower compared to math tracts (paired $t$-test: $p < 0.0001$, $t(17) = 5.72$, Fig. 5a). In the AF, $T_1$ was lower for reading tracts than math tracts when testing within-network connections (paired $t$-test: $p = 0.02$, $t(17) = 2.66$, Fig. 5d), but not when testing connections to the lOTC conjunction fROI (paired $t$-test: $p = 0.13$, $t(16) = 1.58$, Fig. 5g).

Since the SLF and AF are long fascicles, we also evaluated local differences between math and reading tracts across these fascicles. For this, we segmented each fascicle to 100 nodes in each subject and then measured $T_1$ at each node. Examination of the distributions of $T_1$ values across nodes showed lower $T_1$ in the fMWTs of the reading network compared to the math network in the SLF (Fig. 5b) and, to a lesser degree, in the AF (Fig. 5e, h). Examination of $T_1$ values at each node revealed differences across the length of the fascicles. In both the SLF and the AF, $T_1$ differences were more pronounced towards the anterior end of the fascicle, compared to the posterior end (Fig. 5c, f, i).

A similar pattern of results was observed in the right hemisphere (Supplementary Fig. 21), when using constant-size spherical fROIs (Supplementary Fig. 22) and for direct fROI to

fROI tractography (Supplementary Fig. 23). The shorter $T_1$ found along the SLF and AF for tracts associated with reading suggests that these tracts are more heavily myelinated then those associated with math.

## Discussion

Here, we investigated in typical adults what are the shared and dissociated gray and white matter substrates of math (adding) and reading. We found that (i) neighboring gray matter regions in the math and reading networks show similar white matter connectivity, (ii) the AF and SLF support within-network connectivity in both networks, and (iii) within the SLF as well as the AF, tracts associated with math and reading are segregated, and show significant structural differences. Our data thereby open a new avenue of research focused on understanding how sub-bundles within fascicles may contribute to behavior.

Several methodological innovations were key to the present study. First, we used tractography with constrained spherical deconvolution[31], which allowed us to resolve white matter tracts in crossing fiber regions and close to cortex. Second, by combining fMRI and dMRI, and intersecting each subjects' tracts with the GWMI underneath their fROIs, we were able to define the fWMT of the math and the reading networks within individual subjects. Third, we applied qMRI to elucidate structural properties of fascicles involved in math and reading. Prior investigations focused on diffusion measures, such as FA, which show a complex relation to the underlying microstructure of white matter[42,43]. In contrast, $T_1$, measured by qMRI, is correlated with myelin content[35], thereby providing insight about a fundamental microstructural component of these tracts. Fourth, by using identical stimuli for three different tasks that are matched in their working memory load and the amount of verbalization they elicit, we were able to distil cortical regions that are involved in adding, reading or both, while controlling for stimulus differences as well as general cognitive demands.

It should be noted that the fascicles reported here likely do not reflect the entire white matter of math and reading, for three reasons. First, in addition to the connections described here, regions activated during math and reading likely also connect to regions outside the math and reading networks. Second, there are likely additional white matter tracts associated with each region beyond those in fascicles (e.g., U-fibers), which have not been considered here. Third, we focused on addition, and did not investigate neural substrates of other mathematical operations. Previous work has shown that different mathematical operations may vary in their neural substrates[23,44,45]. Particularly, compared to other operations, addition relies more heavily on arithmetic fact retrieval[46] and thus may show more overlap with reading[3]. Future studies can examine which components of the revealed network are specific to addition and which components extend to other mathematical tasks.

Our study yields novel insight on the reading and math networks. First, we show that regions preferentially activated during adding and reading often neighbor (Fig. 1b). Regions involved in math were found to generally be located superior to regions involved in reading. As prior research suggests that white matter development precedes and predicts the location of functional regions involved in reading[47], future developmental research could test if white matter tracts also determine the location of regions involved in math. Second, we show that, even as the SLF and AF are key fascicles for both reading and math, they contain separate sub-bundles for each task. Specifically, analogous to separate lanes on a highway, parallel and segregated tracts within these fascicles are part of either the reading or the math network (Fig. 4, Supplementary Fig. 14-16). These distinct sub-bundles

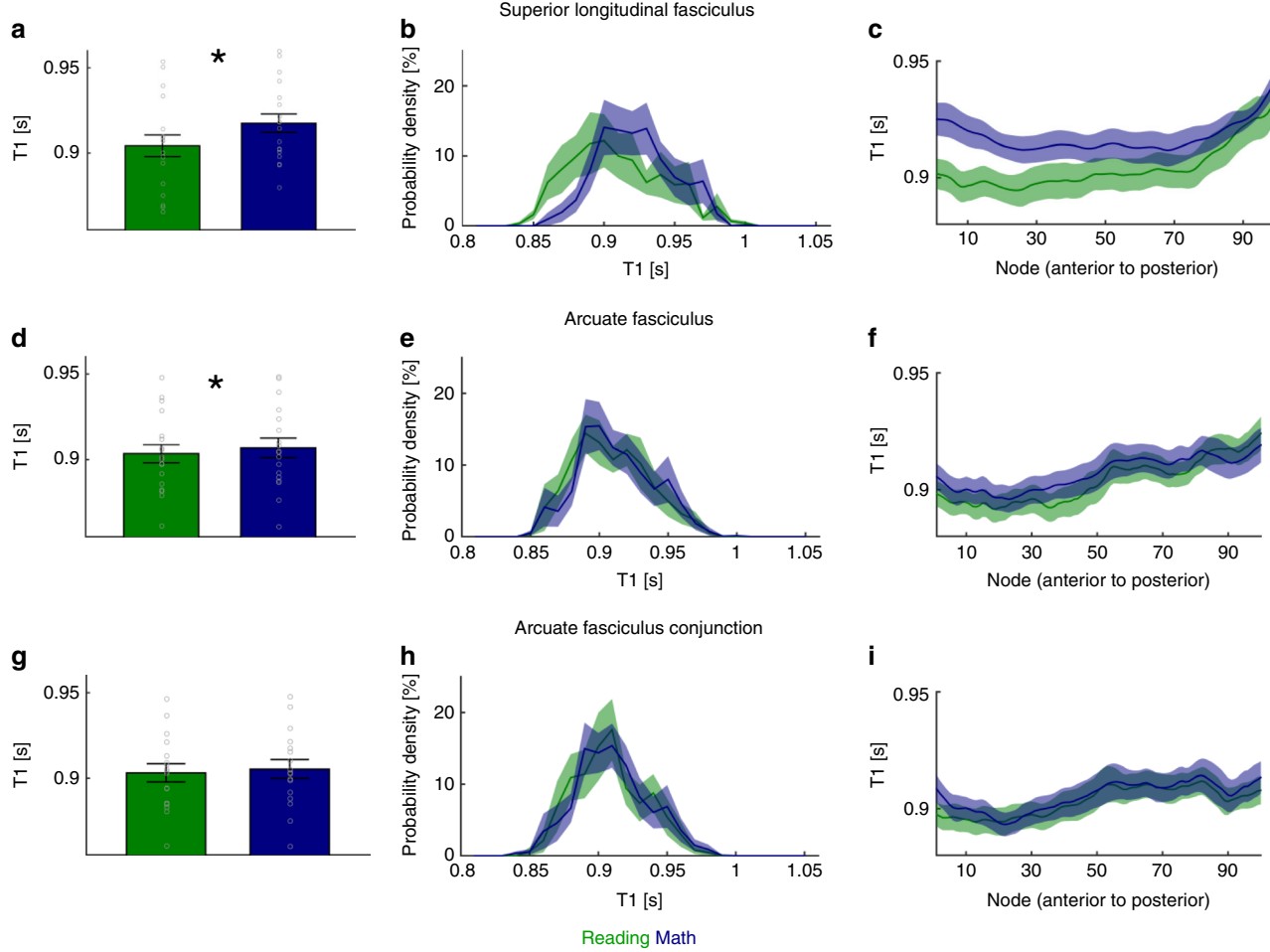

**Fig. 5** Tracts in the SLF and the AF that are associated with reading show shorter proton relaxation time ($T_1$) than those associated with math. **a–c**: $T_1$ measurements for SLF tracts connecting IFG and SMGr in the reading network (green) and PCS and SMGm in the math network (blue). **d–f**: $T_1$ measurements for AF tracts connecting the IFG and STS in the reading network (green) and the PCS and ITG in the math network (blue). **g–i**: $T_1$ measurements for AF tracts connecting the lOTC conjunction fROI with the IFG in the reading network (green) and the PCS in the math network (blue). Left **a**, **d**, **g**: Average $T_1$ for reading- and math-related tracts in the SLF and the AF. Bar graph shows mean across subjects ± SEM. Circles: Individual subjects' data. Asterisk: $T_1$ for math- and reading-related tracts differs significantly, $p < 0.05$ (paired $t$-test). Middle **b**, **e**, **h**: Distribution of $T_1$ values. The histograms show the distribution of $T_1$ values across all nodes in each subject ± SEM. Right **c**, **f**, **i**: Average $T_1$ for reading- and math-related tracts along the SLF and the AF. Line graph shows mean across subjects ± SEM. IFG inferior frontal gyrus, PCS precentral sulcus, SMGr reading fROI in supramarginal gyrus, SMGm math fROI in supramarginal gyrus, STS superior temporal sulcus, ITG inferior temporal gyrus, lOTC lateral occipito-temporal cortex, SLF superior longitudinal fasciculus, AF arcuate fasciculus

suggest that (i) white matter connections of the math and reading networks are more spatially specific than previously thought, and (ii) math and reading are processed largely in parallel in the brain. This, in turn, suggests that improvements in one skill may not translate to the other skill, unless this improvement is linked to broad changes that transcend entire fascicles. Third, strikingly, we found structural differences between math and reading sub-bundles within the SLF and AF. That is, $T_1$ was shorter within reading than math tracts, which suggests more substantial mye-lination of the former than the latter (Fig. 5). Notably, as reading is practiced more frequently and intensely than math during childhood[48], and myelination is dependent on neural activity[49], our findings raise the intriguing possibility that the amount of learning and its resultant neural activity may affect myelination of specific white matter tracts within fascicles.

Our findings make interesting predictions for potential links between white matter and math and reading skills: We hypo-thesize that the properties of the inferior and superior sections of the SLF and AF may independently contribute to reading and

math performance. That is, if myelination improves transmission of information across distributed networks, then $T_1$ of the superior portion of the SLF and AF may correlate with math ability, while $T_1$ of the inferior portions of these fascicles may correlate with reading ability. Accordingly, atypical myelination of tracts within the superior and inferior portion of the SLF and AF during development may also be associated with math or reading learning disabilities, respectively. Future studies with clinical populations can test this hypothesis. Further, we predict that if neural activity promotes myelination, then people with intensive practice in one of these tasks (e.g., ref. [50]) may show lower $T_1$ in the respective tracts compared to lay people. These predictions will be particularly relevant for studies evaluating the efficacy of interventions aimed at improving math and reading skills (e.g., refs. [26,51,52]).

Crucially, the structural differences within the SLF and AF observed in the current study also have implications beyond math and reading. While structural differences within large tracts have recently been shown in the optic radiation, where an

anterior sub-bundle that includes Meyer's loop has higher $T_1$ than the entire optic radiation[53], the present study is the first to reveal such structural differences between fWMT within fascicles. Our data thus encourages a novel research direction that links quantitative properties of functionally-defined sub-bundles to behavior. That is, we believe that understanding the relationship between white matter properties and brain function not only in reading and math, but in a broad range of functions including face processing[54], working memory[55], and attention[56], may be improved if white matter is defined more precisely, by linking it to the specific cortical regions that support each function.

In conclusion, our data show functional and structural segregation of the math and reading networks in the adult human brain. These findings have implications for our understanding of the neural underpinning of math and reading as well as the link between white matter properties and human behavior more broadly.

## Methods

**Participants**. Twenty typical adult participants (10 female, mean age ± SD: 27 ± 6 years, one left-handed) were recruited from Stanford University and surrounding areas and participated in two experimental sessions. Subjects gave their informed written consent and the Stanford Internal Review Board on Human Subjects Research approved all procedures.

**Functional MRI data acquisition and preprocessing**. fMRI data were collected at the Center for Cognitive and Neurobiological Imaging at Stanford University, using a GE 3 tesla Signa Scanner with a 32-channel head coil. We acquired 48 slices covering the entire cortex using a T2*-sensitive gradient echo sequence (resolution: 2.4 × 2.4 × 2.4 mm, TR: 1000 ms, TE: 30 ms, FoV: 192 mm, flip angle: 62°, multiplexing factor of 3). A subset ($N = 12$) of the fMRI data was also used for our previous study[36]. A whole-brain, anatomical volume was also acquired, once for each participant, using a T1-weighted BRAVO pulse sequence (resolution: 1 × 1 × 1 mm, TI = 450 ms, flip angle: 12°, 1 NEX, FoV: 240 mm). The anatomical volume was segmented into gray and white matter using FreeSurfer (version 6.0.0, http://surfer.nmr.mgh.harvard.edu/), with manual corrections using ITKGray (http://web.stanford.edu/group/vista/cgi-bin/wiki/index.php/ItkGray). From this segmentation, each participant's cortical surface was reconstructed. Each participant's anatomical brain volume was used as the common reference space for all analyses, which were always performed in individual native space.

The functional data were analyzed using the mrVista toolbox (http://github.com/vistalab) for Matlab, as in previous work[30]. The data were motion-corrected within and between scans and then manually aligned to the anatomical volume. The manual alignment was optimized using robust multiresolution alignment[57]. No smoothing was applied. The time course of each voxel was high-pass filtered with a 1/20 Hz cutoff and converted to percentage signal change. A design matrix of the experimental conditions was created and convolved with the hemodynamic response function (HRF) implemented in SPM (http://www.fil.ion.ucl.ac.uk/spm) to generate predictors for each experimental condition. Response coefficients (betas) were estimated for each voxel and each predictor using a general linear model (GLM).

**Stimuli and design**. In the fMRI experiment, we presented well-controlled character-like stimuli, which could be used for a reading task, an adding task, and a color memory task (Fig. 1a). These stimuli allowed us to define both reading- and adding-related brain regions within the same experiment, while keeping the visual input constant[30]. At the beginning of each trial, subjects were presented with a cue (Add, Read, or Color), indicating which task they should perform. In the adding task, participants were asked to sum the values of the stimuli and to indicate the correct sum. In the reading task, subjects were instructed to read the word in their head, and to indicate which word had been presented. Finally, in the color task, participants were asked to memorize the color of the stimuli and to indicate which color was shown during the trial. After the cue, four images were shown sequentially, followed by an answer screen. Each image was a morph of a number and a letter. All images in a trial were either number morphs (N, >80% number + < 20% letter) or letter morphs (L, >80% letter + < 20% number), i.e., stimuli that mostly contained information from one category, but held just enough evidence from the other category to be recognizable as both letters and numbers. The same stimuli appeared in all tasks. The answer screen was presented for 2 s and showed the correct answer as well as one incorrect answer at counterbalanced locations left and right of fixation. Participants performed 6 runs, each lasting 6 min; task order was randomized across runs and participants. Prior to the experiment, subjects were given training to ensure that they could perform the task with at least 80% accuracy.

**Participant's performance**. Participants successfully performed all tasks in the experiment (average accuracy (±SE): 88.16(2.43)%). Both accuracy and response times (RTs) differed across the reading, adding, and color tasks (main effect of task: accuracy: F(2,38) = 10.30, $p = 0.0003$, $\eta_p^2 = 0.35$; RTs: F(2,38) = 72.20, $p < 0.0001$, $\eta_p^2 = 0.79$). While accuracy was significantly higher in the reading task, relative to the other two tasks (all ps < 0.002 after Bonferroni correction, n.s. between adding and color), response times were shortest in the adding, intermediate in the reading task, and slowest in the color task (all ps < 0.003 after Bonferroni correction). It is unlikely that performance differences across tasks drove responses across cortex for two reasons: (i) within a task, response accuracy and neural task preference, i.e., the extent of preferential activations for a given task, did not show any clear relationship across participants (parameter maps presented in Supplementary Figs. 1-4 are sorted according to task performance, for group maps see Supplementary Fig. 5), and (ii) accuracy and response times were not consistently different across tasks, yet we could identify functional regions of interest (fROIs) for all tasks (Fig. 1b, Supplementary Figs. 6, 7).

**Functionally-defined gray matter regions**. Reading- and math-related gray matter regions of interest (fROIs) were defined in each participant's cortical surface using both functional and anatomical criteria. For example, for our IFG fROI we took only those voxels that (i) showed the relevant task preference beyond the threshold of T ≥ 3 and (ii) fell within the inferior frontal gyrus. The resulting fROIs were labeled according to their anatomical location. Reading-related fROIs consist of voxels that showed higher responses in the reading than the math and the color task (T ≥ 3, voxel level), while math-related fROIs contain voxels which showed higher responses in the math than the reading and the color task (T ≥ 3, voxel level). We also identified regions that are involved in both math and reading using a conjunction analysis (math > color ∩ reading > color, T ≥ 3, voxel level) and extracted the response profile of the resulting fROIs (Supplementary Fig. 8). For all analyses, we report data from regions that showed a reliable preference for a given task across subjects. That is, we report regions that could be identified in the left hemisphere in at least 90% of the participants. In other words, while in a given individual there may be additional voxels that respond preferentially during reading and/or during adding, here we focus on the most consistent activations across participants.

We found a consistent preference for reading compared to adding and color tasks in four anatomical expanses (example subject in Fig. 1b-green; all subjects in Supplementary Figs. 6, 7): (i) A region in the OTS (left hemisphere: $N = 18$, size ± SE: 492 ± 97 mm³; right hemisphere: $N = 12$, size ± SE: 107 ± 33 mm³). Activations in the OTS were frequently divided into two distinct subregions and we took their union as a single fROI, as we were interested in the large-scale networks associated with reading and math. (ii) A region in the STS, which extended into the middle temporal gyrus (left hemisphere: $N = 20$, size ± SE: 774 ± 184 mm³; right hemisphere: $N = 17$, size ± SE: 343 ± 110 mm³). (iii) A region in the SMG (we refer to this region in the reading network as SMGr) (left hemisphere: $N = 20$, size ± SE: 494 ± 148 mm³; right hemisphere: $N = 19$, size ± SE: 119 ± 34 mm³). (iv) A region in the IFG (left hemisphere: $N = 20$, size ± SE: 1458 ± 255 mm³; right hemisphere: $N = 20$, size ± SE: 668 ± 157 mm³). Activations in the IFG spanned 2–3 clusters and we took their union. Given that reading fROIs were more commonly found in the left hemisphere, the main text focuses on this hemisphere, whereas right hemisphere data is presented in the Supplementary Material (Supplementary Figs. 2, 4, 5, 7, 10, 12, 17, 21).

We also identified four bilateral regions that responded more strongly during the adding task than the reading or color tasks (example subject in Fig. 1b-blue; all subjects in Supplementary Figs. 6, 7): (i) A region in the ITG (left hemisphere: $N = 20$, size ± SE: 680 ± 126 mm³; right hemisphere: $N = 19$, size ± SE: 570 ± 92 mm³). (ii) A region in the IPS (left hemisphere: $N = 19$, size ± SE: 1329 ± 224 mm³; right hemisphere: $N = 18$, size ± SE: 1283 ± 197 mm³). (iii) A region in the SMG (we refer to this region in the math network as SMGm) (left hemisphere: $N = 20$, size ± SE: 893 ± 178 mm³; right hemisphere: $N = 20$, size ± SE: 1170 ± 187 mm³). (iv) A region in the inferior part of the PCS (left hemisphere: $N = 18$, size ± SE: 763 ± 141 mm³; right hemisphere: $N = 19$, size ± SE: 580 ± 100 mm³).

We further tested whether any regions in the brain show higher responses during both the adding and reading tasks compared to the color task (conjunction analysis, adding > color ∩ reading > color; example subject in Fig. 1b-orange; all subjects in Supplementary Figs. 6, 7). Four regions showed a preference for math and reading compared to the color task (i) A region in the IPS (left hemisphere: $N = 18$, size ± SE: 363 ± 90 mm³; right hemisphere: $N = 16$, size ± SE: 251 ± 63 mm³). (ii) A region in the SMG (left hemisphere: $N = 20$, size ± SE: 434 ± 89 mm³; right hemisphere: $N = 17$, size ± SE: 249 ± 54 mm³). (iii) A region in the PCS (left hemisphere: $N = 19$, size ± SE: 350 ± 87 mm³; right hemisphere: $N = 18$, size ± SE: 111 ± 26 mm³). (iv) A region in the lOTC that extended from the ITG to the OTS (left hemisphere: $N = 19$, size ± SE: 720 ± 141 mm³; right hemisphere: $N = 18$, size ± SE: 241 ± 53 mm³).

In addition to math- and reading-related regions, we also defined fROIs involved in color memory. Regions involved in color memory were used to empirically determine the chance level for pairwise connections between any pair of regions in the brain. Color-preferring regions were identified in the medial aspect of the fusiform gyrus and showed significantly higher responses

during the color task than the other two tasks (T ≥ 3, voxel level). These regions were frequently divided into three distinct subregions (likely corresponding to color patches Ac, Cc, and Pc[58]). Given that these subregions are proximal, here we took the union of these color patches (left hemisphere: $N = 19$, size ± SE: $318 \pm 75$ mm³; right hemisphere: $N = 18$, size ± SE: $277 \pm 64$ mm³).

**Diffusion MRI data acquisition and processing.** Diffusion-weighted MRI (dMRI) data were collected in the same participants during a different day than the fMRI data, at the same facility and with the same 32-channel head coil. DMRI was acquired using a dual-spin echo sequence in 96 different directions, 8 non-diffusion-weighted (b = 0) images were collected, 60 slices provided full head coverage (resolution: $2 \times 2 \times 2$ mm, TR: 8000 ms, TE: 93.6 ms, FoV: 220 mm, flip angle: 90°, b: 2000 s mm$^{-2}$).

DMRI data were preprocessed using a combination of tools from MRtrix3[59,60] (github.com/MRtrix3/mrtrix3) and mrDiffusion (http://github.com/vistalab) toolbox (see ref. [61]). We denoised the data using (i) a principal component analysis, (ii) Rician based denoising, and (iii) Gibbs ringing corrections[62–64]. We also corrected for eddy currents and motion using FSL[65] (https://fsl.fmrib.ox.ac.uk/) and we performed bias correction using ANTs[66]. DMRI data were then registered to the average of the non-diffusion-weighted images and aligned to the corresponding high-resolution anatomical brain volume using rigid body transformation. Voxel-wise fiber orientation distributions (FOD) were calculated using constrained spherical deconvolution (CSD)[31] with up to eight spherical harmonics (lmax = 8). These FODs were used for tractography.

**Tractography.** Ensemble tractography[67] was performed using the processed dMRI data and consisted of 3 main steps: We (1) created multiple connectomes that varied in their allowed angle, (2) concatenated these candidate connectomes into one large ensemble connectome, and (3) automatically labelled major fascicles in the ensemble connectome.

(1) Candidate connectome generation: We used MRtrix3[59] (RC3, http://www.mrtrix.org/) to generate five candidate connectomes which varied in the maximum angle (2.25°, 4.5°, 9°, 11.25°, and 13.5°). The goal of this approach was to generate multiple connectomes with tracts with different degrees of curviness, rather than choosing a single connectome with one particular set of parameters[67]. For each connectome, we used probabilistic fiber tracking with the following parameters: algorithm: IFOD1, step size: 0.2 mm, minimum length: 4 mm, maximum length: 200 mm, FOD amplitude stopping criterion: 0.1. We used anatomically constrained tractography (ACT)[68], which utilizes information of different tissue types from the FreeSurfer (https://surfer.nmr.mgh.harvard.edu/) segmentation of each participant's high-resolution anatomical scan to optimize tractography. ACT also allowed us to identify the gray-white-matter-interface (GWMI) directly underneath the fROIs. Seeds for tractography were randomly placed within this interface. This enabled us to focus on those fiber tracts that reach the gray matter. Each candidate connectome consisted of 500,000 streamlines.

(2) The five candidate connectomes were concatenated into one ensemble connectome containing a total of 2,500,000 streamlines using custom Matlab code available in github (https://github.com/VPNL/mrLanes).

(3) We used Automated Fiber Quantification[32] (AFQ, https://github.com/yeatmanlab/AFQ) to segment the ensemble connectome of each participant into 13 well-established major fascicles, most of them are bilateral (Supplementary Fig. 9). The resulting classified connectome was optimized by removing tracts that were located more than four standard deviations away from the mean of their respective fascicle (see e.g., refs. [32,69]).

We conducted all subsequent analyses on these classified white matter tracts within the 13 major fascicles, as we were interested in identifying large-scale, whole-brain networks involved in reading and math.

**Functionally defined white matter tracts (fWMT).** fWMT of each fROI: To identify the tracts associated with each math and reading fROI, we intersected the classified tracts within 13 major fascicles with the GWMI directly adjacent to each of the math- and reading-related fROIs. This yielded the functionally defined white matter tracts (fWMT) of each fROI. We visualized the reading- (Fig. 2a) and math-related (Fig. 2b) fWMT, as well as those tracts that connect to an lOTC region identified in the conjunction analysis (Fig. 2c) in individual subjects. In all plots, tracts are color-coded by the fascicle they belong to; the plots are thresholded at a maximum of 50 tracts per fascicle to enable clearer visualization. We also quantified the distribution of the fWMT of each fROI across the 13 main fascicles (Fig. 2a, b, c-middle). Finally, we created a schematic representation of the fascicles each fROI connects to, where the width of the line represents the proportion of the total fWMT occupied by that fascicle (Fig. 2a, b, c-right).

Within-network connections: To evaluate the tracts associated with the math and reading networks, we identified pairwise connections between fROIs within each network. We intersected the fWMT of each fROI with the GWMI underneath each of the other fROIs of the same network (either math or reading) to identify tracts that connect to at least two fROIs of the network. For both reading (Fig. 3a–d) and math (Fig. 3e–h), we first visualized all pairwise fWMT in each individual

subject, color coded by the fascicle they belong to (Fig. 3-a, e show a representative subject). Next, we quantified the pairwise connections using the dice coefficient[41] (DC; Fig. 3-b, f):

$$DC = \frac{2(A \cap B)}{A + B}$$

where A represents all tracts that connect to one fROI, B represents all tracts that connect to the second fROI, and A ∩ B represents those tracts that connect to both fROIs. The DC quantifies the similarity of two samples; a DC of 1 indicates complete overlap (i.e., each tract that connects to one fROI, also connects to the other fROI), while a DC of 0 indicates that there are no tracts that connect to both fROIs. We also estimated chance level DCs, by calculating the DC for pairwise fWMT between each of the fROIs of the math and the reading network to ventral regions activated during the color memory task (pairwise connections with the OTS were not included, due to its close anatomical proximity with fROIs involved in processing color). The average DCs of these pairwise connections were used as chance level DCs for each network, given that we expected connections to color-related regions to be irrelevant for participants' math and reading skills. In addition, we also evaluated what percentage of the pairwise connections belong to each fascicle (Fig. 3-c, g). Finally, we created a schematic representation of these pairwise connections, where the width of the lines is determined from the DC and the color of the lines indicates which fascicle contributed most strongly to this connection (Fig. 3-d, h).

Between-network connections: We also tested for between-network connections, using two approaches: (i) We identified and quantified the pairwise connections between the lOTC conjunction fROI and each non-neighboring fROIs in both the math and reading network with the methods described above (Fig. 3i–l), and (ii) we identified connections between pairs of non-neighboring fROIs across networks using the same methods as above except that in each pair one fROI was from the reading network and the other fROI from the math network (Fig. 3m–o).

**Quantification of segregation within fascicles.** For those fascicles that contributed to significant pairwise connections in both the math and the reading network, we tested whether fWMT within these fascicles are segregated by network. Specifically, we evaluated if (i) within the SLF, tracts connecting SMGr to the IFG in the reading network are intertwined or segregated from tracts connecting SMGm to the PCS in the math network (Fig. 4a–c, tracts that connect to all 4 fROIs were excluded), (ii) within the AF, tracts connecting the STS and the IFG are intertwined or segregated from tracts connecting the ITG and the PCS (Fig. 4d–f, tracts that connect to all 4 fROIs were excluded), and (iii) within the AF, tracts connecting the lOTC conjunction fROI with the IFG are intertwined or segregated from tracts connecting the same lOTC fROI with the PCS (Fig. 4g–i, tracts that connect to all 3 fROIs were excluded).

For this, first we visualized pairwise connections of the math and reading network within each fascicle, in each individual subject, and visually inspected their spatial layout (Fig. 4-a, d, g and Supplementary Figs. 14-16). Then, we resampled each tract in each subject to 100 equally spaced nodes (i.e., locations) between the way-point ROIs used by AFQ to define the fascicle. This procedure ensured that we have the same number of measurements per subject, even though the absolute length of the fascicles may vary across subjects. Finally, we quantified the segregation of fWMT of each network within each fascicle using two complimentary approaches: (1) we measured the distance of each tract from the core tract of within-network-tracts, as well as, its distance from the core tract of the other network; then we tested whether the former is lower than the latter and (2) we used an independent classifier to test if across nodes fWMT can be identified as belonging to either the reading or the math network based on their anatomical location.

(1) Distance to core tract within and between networks: We first calculated the core (mean) tract of the pairwise connections within the SLF and AF, separately for math and reading-related fWMT, using AFQ. Next, we measured, within each subject and at each node, how far away (Euclidian distance in millimeter, derived from x,y,z coordinates) each tract is from the core tract within its network and the core tract of the other network (Fig. 4-b, e, h). We expected tracts to be closer to the core tract of their own network if math- and reading-related fWMT are segregated within the fascicle, but equal distant from both core tracts if math- and reading-related fWMT are intertwined within the fascicle.

(2) Classification: We tested if across the length of the fascicle we can classify tracts as math or reading tracts based on their anatomical location. At each node and within each subject, the coordinates of all math and reading-related tracts were used to train a linear support vector machine (SVM) classifier. The SVM from each node was used to classify tracts at the fifths more posterior node (we chose the fifths node rather than a neighboring node to ensure independence of training and test data) as either math or reading tracts (Fig. 4-c, f, i). We expected the classifier to perform at chance (50% accuracy) if math and reading-related tracts are intertwined, but significantly above chance if math and reading tracts are spatially segregated across the lengths of the fascicle.

**Quantitative MRI data acquisition and preprocessing.** Quantitative MRI (qMRI[33]) data were collected within the same session and with the same head coil

as the dMRI data. $T_1$ relaxation times were measured from four spoiled gradient echo images with flip angles of 4°, 10°, 20°, and 30° (TR: 14 ms, TE: 2.4 ms). The resolution of these images was later resampled from $0.8 \times 0.8 \times 1.0$ mm$^3$ to 1 mm isotropic voxels, and qMRI data was aligned with the high-resolution anatomical scan using rigid body transformation. We also collected four additional spin echo inversion recovery (SEIR) scans with an echo planar imaging read-out, a slab inversion pulse and spectral spatial fat suppression (TR: 3 s, resolution: $2 \times 2 \times 4$ mm, 4 echo time set to minimum full, 2× acceleration, inversion times: 50, 400, 1200, and 2400 ms). The purpose of these SEIRs was to remove field inhomogeneities.

Both the spoiled gradient echo and the SEIR scans were processed using the mrQ software package (https://github.com/mezera/mrQ) for Matlab to estimate the proton relaxation time ($T_1$) and macromolecular tissue volume (MTV) in each voxel, as in previous studies[33,70]. The mrQ analysis pipeline corrects for RF coil bias using the SEIRs scans, which produces accurate proton density (PD) and $T_1$ fits across the brain. MrQ also produces maps of MTV, by calculating the fraction of a voxel that is non-water.

**Comparison of $T_1$ for math and reading tracts**. We used the $T_1$ maps to evaluate tissue properties of tracts of the math or the reading network (Fig. 5; MTV data are presented in Supplementary Fig. 20). We focused on those fascicles that had the largest contribution to significant within-network connectivity in both networks. Within the SLF, we compared tracts connecting SMGr to the IFG in the reading network with tracts connecting SMGm to the PCS in the math network (Fig. 5a-c, tracts that connect to all 4 fROIs were excluded). Within the AF, we compared tracts connecting the STS and the IFG in the reading network with tracts connecting the ITG and the PCS in the math network (Fig. 5d-f, tracts that connect to all 4 fROIs were excluded). Within the AF, we also compared tracts connecting the lOTC conjunction fROI with the IFG in the reading network to those connecting the lOTC conjunction fROI to the PCS in the math network (Fig. 5g-i). We first evaluated the mean $T_1$ values of each tract in each subject and tested if there are between-network differences (Fig. 5-a, d, g). Then, we resampled the tracts to 100 equally spaced nodes in-between the way-point ROIs used by AFQ to identify the fascicle. We visualized the distribution of $T_1$ values, using data from all nodes and subjects, and compared the distribution between math and reading tracts (Fig. 5-b, e, h). Finally, we visualized the $T_1$ of math and reading tracts across the different nodes, to determine if $T_1$ differences are homogenous or heterogeneous across the length of the fascicle (Fig. 5-c, f, i).

**Control analyses**. In addition to our main approach, we conducted two types of control analyses:

(1) Constant-sized fROI. We replicated our analyses using constant-size spherical ROIs centered on our fROIs (radius = 7 mm, this radius was chosen based on previous studies[17,29]). The goal of this control was to test if differences in fROI size across participants and regions influenced the identified math and reading networks. Results of these analyses are presented in the Supplementary Materials (Supplementary Figs. 11, 13, 18, 22).

(2) Direct fROI-to-fROI tractography. We replicated the pairwise connections between fROIs analyzed in Figs. 4, 5 by tracking between fROIs. That is, the GWMI underneath the anterior fROI in each pair was used as a seed and the GWMI of the other fROI was used as a target for tractography. The advantage of this approach is that it allows to better control the number of tracts seeded in the GWMI of each fROI. However, in contrast to our main approach, this control analysis only tracks between pairs of fROI and hence does not provide a picture of the entire math and reading networks. We used the following parameters for this control analysis: algorithm: IFOD1 with ACT, lmax: 8, step size: 0.2 mm, minimum length: 4 mm, maximum length: 200 mm, FOD amplitude stopping criterion: 0.1, angle: 13.5°. We continued tracking between the fROIs until (i) we found 100 tracts or (ii) we attempted 100.000 times to seed tracts. Tracts were classified and cleaned using AFQ as described above.

**Statistics**. Repeated measures analyses of variance (ANOVAs) were used to test for accuracy and RT differences across tasks, and to test if fROIs identified in the conjunction analysis respond equally strongly to math and reading tasks. We used paired $t$-tests to evaluate if $DCs$ and decoding accuracies differed significantly from chance. We also used paired $t$-test to evaluate if there are differences in Euclidian distance of fWMTs to the within- and between-network core tracts and if there are $T_1$ differences between math and reading-related tracts. When more than one $t$-test was conducted, the statistical threshold was Bonferroni-adjusted to account for multiple comparisons.

**Reporting summary**. Further information on research design is available in the Nature Research Reporting Summary linked to this article.

## Data availability

The data generated in this study will be made available upon reasonable request. Source data for Figs. 2–5 are made available in github (https://github.com/VPNL/mrLanes). A reporting summary for this Article is available as a Supplementary Information file.

## Code availability

The fMRI and qMRI data were analyzed using the open source mrVista software (available in GitHub: http://github.com/vistalab) and mrQ software (available in GitHub: https://github.com/mezera/mrQ) packages, respectively. The dMRI data were analyzed using open source software, including MRtrix3[59] (http://www.mrtrix.org/) and AFQ[32] (https://github.com/yeatmanlab/AFQ). We make the entire pipeline freely available; custom code for preprocessing, tractography and further analyses are available in github (https://github.com/vistalab/RTP-preproc; https://github.com/vistalab/RTP-pipeline; https://github.com/VPNL/mrLanes). Code for reproducing all figures and statistics are made available in github as well (https://github.com/VPNL/mrLanes).

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

## Acknowledgements

This research was supported by the National Institute of Health (NIH; 1R01EY023915), by the Deutsche Forschungsgemeinschaft (DFG; GR 4850/1–1) and by an Innovation Grant from the Stanford Center for Cognitive and Neurobiological Imaging (CNI). The authors would like to thank Brianna Jeska for her help with the data collection.

## Author Contribution

M.G. and K.G.S. designed the study. M.G. collected the data. M.G., Z.Z., and G.L.U. developed code used for functional, diffusion and quantitative data analyses. M.G., G.L.U., and K.G.S. analyzed the data. M.G. and K.G.S. wrote the manuscript.

## Additional information

**Competing interests:** The authors declare no competing interests.

