## [Peer Review File · Nature Communications]

Reviewers' Comments:

Reviewer #1:

Remarks to the Author:

The present study used a multimodal imaging approach to examine the shared and distinct white matter tracts that connect areas that are relevant for reading and arithmetic. The authors functionally defined regions via fMRI in 14 subjects and then examined which tracts are shared and distinct for reading and arithmetic. Their findings revealed that both the superior longitudinal fasciculus (SLF) and arcuate fasciculus (AF) are shared between reading and arithmetic. Follow-up analyses indicate that within these fasciculi, some parts are related to reading and some to arithmetic, suggesting some degree of specificity.

There are several things I like about this study. The authors use fMRI to functionally define the reading and arithmetic-related regions and subsequently use these to investigate white matter tracts. I also appreciated the authors use of spherical deconvolution to deal with the crossing fibers problem in classic DTI. Such approaches have been applied in reading research, but to my knowledge they have not been used in studies on arithmetic or mathematics. The authors use state-of-the-art techniques to use these white matter data, for which they are commended. On the other hand, there are some major issues that prevent me from being positive about the manuscript. These deal with the sample size, the way they analyzed the fMRI data and the imho to far-fetched implications the authors derive from the current data (to development and to understanding co-morbidity). In addition, I have some minor points which the authors might want to consider in a revision of the manuscript.

MAJOR POINTS

The sample size of this study ($n = 14$) is too small to draw strong conclusions from the current data. At maximum, I would consider these data as being preliminary and promising. If the authors would add more subjects to their sample – the sample of typically developing adults is not too arduous to recruit – they would be in a much better position to draw conclusions from the current data.

I really like the authors approach to use fMRI localizer tasks to define their ROIs, yet I think they miss an opportunity to really answer their question of overlap vs. distinctness through the way they selected their functional contrasts. Specifically, the authors use the reading vs. other tasks (including arithmetic) and arithmetic vs. other tasks (including reading) to define their ROIs, but these result in areas that are either unique to reading or unique to arithmetic. This is feasible if they want to focus on differences between reading and arithmetic, but not on their communality. If they want to say something about the latter, they should in fact look at the conjunction between the reading and arithmetic localizer. The authors have all the data to do this. This results in a set of ROIs that reflect the communality of reading and arithmetic, and the authors can then investigate which white matter tracts connect these overlapping regions.

In their introduction, the authors suggest that there is no research on the white matter correlates of mathematics (page 4). I agree that there are not so many studies, yet there are quite some that have used DTI and VBM, which are summarized in this systematic reviews by Matejko & Ansari (2015, *Neurosc. And Biobehav. Rev.*) and Peters & De Smedt (2018, *Dev. Cogn. Neurosc.*), and this should be credited.

In their analysis of the data, the authors report data for reading in the left hemisphere only and they do not further investigate the right hemispheric data, despite the fact that significant activation clusters on the right were observed. I think this is not correct, and they should also examine these right hemispheric connections. It was also not clear whether or not they investigated the right

hemisphere for arithmetic, but I assumed they did so. These right hemispheric ROIs should all be explicitly discussed in the manuscript.

It is unclear why the authors segmented the SLF and AF in 30 nodes. Why 30? They should justify why they used this type of segmentation.

The paper draws suggests numerous implications that are too overstated given the current data. Firstly, the authors indicate that their paper reveals something about comorbidity of learning disorders. This is not at all the case: they collected their data in typically developing individuals and the participants under study are adults (who have developed these abilities) but not children. As a result, this paper cannot really tell something about development or about comorbidity. Second, the authors claim that their findings provide evidence for the triple-code model. This is not the case. For many reasons. One is that the triple-code model is about how we process numbers in different codes (verbal, visual, analogue) but this is not what the authors investigated. Instead they focused on calculation, and not on number processing per se. It is true that different types of calculation, i.e. different operations, could be linked indirectly to the different codes. But as the authors only examined one type of operation (which may be the closest to the verbal code), their data cannot speak to this issue.

The authors highlight that segmenting large white matter tracts into sub-bundles is important to further our theoretical understanding of how networks are supported by white matter tracts. I agree, but they should clarify what these segments can tell us about these networks. How should we interpret them?

MINOR ISSUES

Page 3, the authors suggest reasons for overlap between reading and math. One area of overlap, which can be explicitly linked to the arcuate fasciculus as well as activity in temporoparietal cortex is the hypothesis that arithmetic facts are phonologically represented, and that this might represent an overlap between reading and arithmetic and might explain why reading and arithmetic correlate and why their disorders co-occur. Many studies have investigated this (including neuroimaging work) and I feel this work should be credited.

The authors indicate that their reading and math task are matched. I do not fully agree, since the math task is only single-digit addition and is, to a lesser extent, reflective of mathematical processing per se. On a related note, the authors focus in their work on arithmetic and not mathematics, the latter is a much broader ability.

Were all subjects right-handed? Please clarify.

Please provide more details on the motion assessment of the fMRI data. Add the criteria and report how many there were discarded.

Reviewer #2:

Remarks to the Author:

The current manuscript attempts to map the white matter fascicles involved in reading and mathematical processing. The authors used a task that they very recently developed to identify cortical fROIs associated with reading and math, and used AFQ to identify 12 major fascicles. They next dilated the tract endpoints by 7mm, and the resulting tracts that intersected with any fROI were

designated to "functionally-defined white matter tracts." The authors report the fascicles that each domain is associated with, and that the individual tractography streamlines are segregated within these fascicles.

My largest concern is that the approach used here relies on an enormous leap of faith in the assumption that (1) a region's proximity to a fascicle endpoint indicates that its efferent or afferent axons travel through that nearby fascicle, and (2) that tracts only enter and exit a fascicle at its endpoints. The authors would need to provide a substantial degree of prior anatomical evidence to show that all gray matter within 7mm of a fascicle terminus are connected to it, and that axons always enter or exit a fascicle at its terminus. Otherwise, the authors need to directly seed the individual gray matter fROIs for tractography, which a very large corpus of prior work has done. This will be extremely important in order to know what regions each fROI is connected to, as well as which fascicles are used to connect them.

Other methodological concerns:

You report that you acquired data from 14 subjects, but most figures and text refer to 12 subjects. Why were 2 subjects excluded?

Please also include subject handedness.

At the end of pg 24: "we report data from regions that showed a reliable preference across participants" and "here we focus on the most robust activations." Please provide an operational definition that delineates exactly how you chose these 8 fROIs.

Also on pg 24: "regions were defined in each participant's cortical surface using both functional and anatomical criteria." What were those anatomical criteria?

Please list the 12 fascicles that you initially identified with AFQ. It would also be helpful to include a supplementary figure that illustrates each of them.

For the svm analysis: why did you use a 2nd degree polynomial instead of linear? What were your other hyperparameters? How were they chosen/optimized?

Since your analyses are based on fROI-fascicle endpoint distance, they will be sensitive to inter-ROI distance. What are the distance minima between each neighboring fROI (ie IFG/PCS, STS/ITG/OTS, SMG ROIs) within the volume?

You do not specify what MR-spaces each analysis was performed in. Was dMRI data transformed to anatomical? Was tractography performed in native dMRI space? Were fROIs transformed to anatomical? How did you perform all of these registrations and transformations?

Results:

Please also include accuracy & RT for each subject, as well as whole brain contrast maps (as heatmaps as opposed to binary labels) for each subject, matched in order as the aforementioned behavioral data

The fROIs appear to be extremely variable in size across regions and subjects. fROI size will unquestionably affect the likelihood that a dilated fascicle endpoint will intersect that fROI. If you are using anatomical constraints, it would be best to use a percentile-based threshold (e.g. top 10%) rather than an absolute threshold ($t=3$), which will guarantee similar sized fROIs across subjects (and fROIs if the anatomical constraints are the same size).

Please provide a group-average heatmap of math and reading responses, which will help the reader evaluate the reliability of these regions across subjects.

For the analyses in Fig3: this would be more clearly interpretable in the context of between vs within tasks, as opposed to simply within task. Please repeat this analysis for math->reading fROIs as well. In any case, since the question here is whether math/reading fROIs preferentially connect to other math/reading fROIs, and not whether they are both 7mm away from coinciding fascicular endpoints, this would be much better if each region was a seed and target for each other region. Currently, high DC just means that both regions are near the endpoints of fascicle, and not that they

are connected to each other.

For the analyses in Fig4, the simplest thing to do would be to identify all voxels in each tract group, and compute an overlap metric such as DICE over the sets of voxels occupied by each group of tracts. Then test against the null hypothesis that they overlap completely ($DC = 1$) or against an empirical null distribution via permutation, resubstitution, or something similar. The metrics used currently are indeed interesting but rather complicated, and a more parsimonious test of overlap would be preferred.

Lastly, these subjects are undeniably identical with those from the same authors' 2018 NeuroImage paper, both in response patterns, and in cortical folding and surface topography. In that paper, there were 15 subjects. Why was one subject excluded?

I greatly hope that the authors do not simply resubmit this manuscript elsewhere as-is, and take time to re-evaluate the analyses that they employed to answer the questions they are attempting to address.

Reviewer #3:

Remarks to the Author:

The authors report on an innovative study using a multimodal approach involving first functional MRI for locating activity related to math and reading, followed by diffusion MRI and quantitative MRI to define their connected fascicles. The study is technically strong, the findings are interesting, and the manuscript is suitable to a wide readership as it presents a major advance in understanding subdivisions of fascicles. There are several areas where clarification is needed.

Main Concerns:

The authors refer to arithmetic, yet the task involves only addition. It has been proposed that addition (and multiplication) of small numbers is more similar to reading in terms of overlap (due to verbal retrieval mechanisms used for both) and that subtraction (and division) is different from these, relying more on procedural-based computation. It seems therefore that the manuscript needs to mention that the findings are specific to addition and not arithmetic at large.

Given the overlap between reading and addition, it seems strange that the investigation focused on read>math and vice versa. What about areas where function is reported for both? It would be critical to examine the fibers connected to regions that show functional overlap (conjunction) for math and reading and test if they are segregated or intertwined. Without this information we still do not have the full picture of how these two skills are shared versus separate. By only looking at areas that are functionally segregated, there seems to be a greater likelihood that they are also anatomically segregated.

The manuscript does not report on the performance (accuracy and reaction time) of the fMRI task. Are the three tasks matched on performance? One concern is that the reading task is hard (it is difficult to read the "letters"). Another is that the memory load on the reading and math tasks are different: the reading task relies more on short term memory while the math task relies on working memory because of the manipulation performed during addition.

Minor Concerns:

Abstract. The abstract is very broad and would benefit from removal of the broader claims at the beginning and the end and instead provide more details on the details of the study.

Introduction. "The VWFA is thought to process visually presented words^{18,19} and is causally involved in word recognition, as lesioning it produces dyslexia²⁰ " Better to use the term acquired dyslexia or alexia as dyslexia usually refers to developmental dyslexia.

Para beginning with "To date, it is unknown if these fascicles, which are important for reading, are also associated with mathematical processing." This needs to be preceded by information on how reading and math are connected. Without it, it may seem odd to the reader why the same fascicles would be associated with reading and math.

"Third, to our knowledge, no study has examined the relationship between gray and white matter substrates of the math network. That is, it is unknown which fascicles intersect with cortical regions involved in mathematical processing." this statement does not seem to fit with the two goals that follow – as they do not describe cortical regions, just white matter. Please also note prior work referenced below.

Methods. Were the scans conducted on the same day?

If I understand it correctly, the scans did not cover the entire cortex- which areas were omitted?

Discussion. The discussion makes many different points using numerical text features. This feels unnecessary. For example, the authors should identify the main findings (two to three) and then turn to a broader discussion which includes other information, without trying to highlight each as a major achievement. Similarly, the fact that the results align with expectations set up by the triple code is highlighted as an important finding, but it seems that the results of a study on arithmetic should align with the most prominent model of math processing. It would be awkward if they did not, so this observation could be toned down. Likewise, there seems to be a strong claim of novelty all around which is not necessary since the aspects that are novel are clear; overdoing it may give the appearance that some prior work has been overlooked. For example, the authors should probably acknowledge the following study using functional seed regions for fiber tracking:

Klein, E., Moeller, K., Glauche, V., Weiller, C., & Willmes, K. (2013). Processing pathways in mental arithmetic-evidence from probabilistic fiber tracking. *PLoS One*, 8(1), e55455.

<https://doi.org/10.1371/journal.pone.0055455>

Generally speaking the approach is highly innovative and the findings are interesting. Additional reporting on areas where reading and addition show overlap in terms of activity would strengthen the manuscript.

Point by point answers to reviewer comments are indicated in green below each comment.

Reviewer #1 (Remarks to the Author):

The present study used a multimodal imaging approach to examine the shared and distinct white matter tracts that connect areas that are relevant for reading and arithmetic. The authors functionally defined regions via fMRI in 14 subjects and then examined which tracts are shared and distinct for reading and arithmetic. Their findings revealed that both the superior longitudinal fasciculus (SLF) and arcuate fasciculus (AF) are shared between reading and arithmetic. Follow-up analyses indicate that within these fasciculi, some parts are related to reading and some to arithmetic, suggesting some degree of specificity.

There are several things I like about this study. The authors use fMRI to functionally define the reading and arithmetic-related regions and subsequently use these to investigate white matter tracts. I also appreciated the authors use of spherical deconvolution to deal with the crossing fibers problem in classic DTI. Such approaches have been applied in reading research, but to my knowledge they have not been used in studies on arithmetic or mathematics. The authors use state-of-the-art techniques to use these white matter data, for which they are commended. On the other hand, there are some major issues that prevent me from being positive about the manuscript. These deal with the sample size, the way they analyzed the fMRI data and the imho to far-fetched implications the authors derive from the current data (to development and to understanding co-morbidity). In addition, I have some minor points which the authors might want to consider in a revision of the manuscript.

MAJOR POINTS

The sample size of this study ($n = 14$) is too small to draw strong conclusions from the current data. At maximum, I would consider these data as being preliminary and promising. If the authors would add more subjects to their sample – the sample of typically developing adults is not too arduous to recruit – they would be in a much better position to draw conclusions from the current data.

Thank you for your comment. In line with your suggestion, we recruited additional subjects and increased our sample size from $N=14$ to $N=20$. A sample size of 20 is frequently used in the field of the current study and we are hopeful that you will find this satisfactory. We would also like to provide additional justification for our choice of sample size:

1) In our previous fMRI study (Grotheer et al. 2018) we used a sample size of $N=15$ to define regions of the math and reading networks and also validated the response preferences of these regions in a split-half analysis of the data. This suggests that a sample size of at least 15 participants is sufficient to reliably identify math and reading related regions with fMRI.

2) We now also performed power analyses (in G*Power, <https://libguides.library.kent.edu/statconsulting/GPower>, a program which uses mean and SD to estimate the required sample size) on our key effects in the dMRI and qMRI data. First, the segregation of math and reading tracts within the SLF, quantified by training a linear SVM to distinguish math tracts from reading tracts based on their anatomical location (Fig. 4c), requires a sample size of $N=3$. Second, the T1 differences between the math and reading related tracts in the SLF (Fig. 5a) requires a sample size of $N=5$. In the current version of the manuscript we now included 4 times as many subjects.

3) Finally, we would like to highlight that our approach differs from the common practice in the field that uses group analyses, which often require larger N s to find significant effects. Here, we used individual subject analyses, analyzing data in each subject's native brain space, an approach that is more work intensive, but improves sensitivity, spatial precision, and reproducibility.

I really like the authors approach to use fMRI localizer tasks to define their ROIs, yet I think they miss an opportunity to really answer their question of overlap vs. distinctness through the way they selected their functional contrasts. Specifically, the authors use the reading vs. other tasks (including arithmetic) and arithmetic vs. other tasks (including reading) to define their ROIs, but these result in areas that are either unique to reading or unique to arithmetic. This is feasible if they want to focus on differences between reading and arithmetic, but not on their communality. If they want to say something about the latter, they should in fact look at the conjunction between the reading and arithmetic localizer. The authors have all the data to do this. This results in a set of ROIs that reflect the communality of reading and arithmetic, and the authors can then investigate which white matter tracts connect these overlapping regions.

Thank you for suggesting this additional analysis. We liked this idea and added the conjunction analysis to our manuscript (see **Figs 2-5**). The conjunction analysis revealed several regions in the brain (in the IPS, SMG, PCS, and IOTC) that are more strongly activated during math and reading tasks than in the color task (**Supplementary Fig. 6-7**). Interestingly, the regions detected in the conjunction analysis substantially overlap with the math network. In fact, a direct comparison of responses to math and reading in these regions shows that while they are activated in both tasks, they are activated more strongly during the math than the reading task (**Supplementary Fig. 8**). The only exception was a cluster in the lateral occipito-temporal cortex (IOTC), which showed no preference for either task. Indeed, the IOTC overlapped with both a region in the ITG which is involved in math as well as a region in the OTS which is involved in reading (**Supplementary Fig. 6-7**). Given that the other fROIs overlap with the math network, to examine additional connections that were not included in our prior analyses and may be involved in common processing to both math and reading, we evaluated the white matter connections of the IOTC fROI in subsequent analyses, results shown in **Fig. 2-5**. Critically, even though the IOTC is involved in both math and reading, tracts within the AF that connect the IOTC to the PCS in the math network remain segregated from those that connect the IOTC to the IFG in the reading network (**Fig. 4**). This strengthens our conclusion about parallel processing streams for math and reading.

In the results section we now write:

On page 7: “We also considered regions involved in both math and reading, defined by the conjunction of both higher responses during math than color as well as during reading than color task (conjunction analysis, $math > color \cap reading > color$).”

On page 10: “Four regions in the brain showed higher responses during both the adding and reading tasks compared to the color task (conjunction analysis, $math > color \cap reading > color$; **Fig. 1b-orange, Supplementary Fig. 6-7**): (i) A region in the intra-parietal sulcus (IPS; left hemisphere: $N=18$, $size \pm SE: 363 \pm 90 \text{ mm}^3$; right hemisphere: $N=16$, $size \pm SE: 251 \pm 63 \text{ mm}^3$). (ii) A region in the SMG (SMG, left hemisphere: $N=20$, $size \pm SE: 434 \pm 89 \text{ mm}^3$; right hemisphere: $N=17$, $size \pm SE: 249 \pm 54 \text{ mm}^3$). (iii) A region in the inferior part of the precentral sulcus (PCS; left hemisphere: $N=19$, $size \pm SE: 350 \pm 87 \text{ mm}^3$; right hemisphere: $N=18$, $size \pm SE: 111 \pm 26 \text{ mm}^3$). (iv) A region in the lateral occipito-temporal cortex (IOTC) that extended from the ITG to the OTS (left hemisphere: $N=19$, $size \pm SE: 720 \pm 141 \text{ mm}^3$; right hemisphere: $N=18$, $size \pm SE: 241 \pm 53 \text{ mm}^3$). Except for the IOTC region, these fROIs were small and tended to overlap with the math fROIs (**Fig 1b, Supplementary Fig. 6-7**). Indeed, responses in the IPS, PCS, and SMG conjunction fROIs were significantly stronger during the adding task than the reading task (**Supplementary Fig. 8**; ANOVA with hemisphere, task and stimulus as factors; main effect of task: IPS: $F(1,14)=17.30$, $p=0.001$, $\eta^2=0.55$; PCS: $F(1,16)=12.97$, $p=0.002$, $\eta^2=0.45$; SMG: $F(1,16)=19.37$, $p=0.0004$, $\eta^2=0.55$). The only conjunction fROI in which responses during adding and reading did not differ significantly was the IOTC (main effect of task: $F(1,17)=3.83$, $p=0.07$, $\eta^2=0.18$). This region overlaps with regions in both the math network (in the ITG) and the reading network (in the OTS). Thus, in subsequent analyses we considered the IPS, PCS, and SMG regions as part of the math network, but the IOTC as a conjunction region involved in both tasks.”

In their introduction, the authors suggest that there is no research on the white matter correlates of mathematics (page 4). I agree that there are not so many studies, yet there are quite some that have used DTI and VBM, which are summarized in this systematic reviews by Matejko & Ansari (2015, *Neurosc. And Biobehav. Rev.*) and Peters & De Smedt (2018, *Dev. Cogn. Neurosc.*), and this should be credited.

Thank you for highlighting these relevant contributions. To credit previous research, we have rephrased the introduction, on page 5 we now write: *“Although it has been suggested that math and reading utilize several overlapping cognitive processes⁴, presently it is unclear if fascicles associated with reading, are also associated with mathematical processing. This fundamental gap in knowledge is due to four main reasons: First, substantially more neuroscience research has been done on the neural bases of reading^{4,5} than the neural bases of math⁶. Second, most prior studies have evaluated either the neural bases of math^{7,24–27} or the neural bases reading^{18,21,28–33}, but not both systems within the same individuals (for an exception see²). Third, no study directly examined the white matter connections associated with functional regions involved in mathematical processing within the same individuals. Previous work on the neural substrates of mathematical processing has focused on linking gray matter structure, evaluated with voxel-based morphometry, and white matter (³⁴, for review see^{6,7}). The only study that evaluated the white matter connections of functional regions related to mathematical processing evaluated gray and white matter in distinct groups of participants³⁵. While this study provided first insight on the white matter connections of the math network, more work is needed to understand these connections within the same individuals. In contrast, a few previous studies have examined which white matter fascicles connect to a key cortical region of the reading network, namely the VWFA, evaluating functional regions and white matter connections within the same subjects^{18–20}. Thus, the goal of this study is twofold: (1) identify and quantify the white matter connections of cortical regions involved in mathematical processing within individual subjects and (2) determine which aspects of this white matter are unique to the math network, and which are shared with the reading network.”*

In their analysis of the data, the authors report data for reading in the left hemisphere only and they do not further investigate the right hemispheric data, despite the fact that significant activation clusters on the right were observed. I think this is not correct, and they should also examine these right hemispheric connections. It was also not clear whether or not they investigated the right hemisphere for arithmetic, but I assumed they did so. These right hemispheric ROIs should all be explicitly discussed in the manuscript.

Thank you for your comment. In the original version of the manuscript, we had presented right hemisphere data for the math network in the supplementary materials, but did not present right hemisphere data for the reading network. We did not present right hemisphere data for reading, as reading fROIs could be identified in less than half of the subjects in the right hemisphere. This observation is well in line with previous work showing that the reading network is left lateralized (e.g., Behrmann and Plaut, *Ann. N.Y. Acad. Sci.*, 2015). This also resulted in small Ns for the analysis of pairwise white matter connections between the regions of the reading network in the right hemisphere. Because the goal of the paper is to compare white matter connections of the math and reading networks within the same individuals, we previously focused on the left hemisphere, in which we could identified fROIs of both networks in the majority of participants.

As we now obtained data from a larger number of participants (N=20), even with only half the subjects showing activations for reading in the right hemisphere, there is now a sufficient number of subjects to show meaningful results for both reading and math in the right hemisphere. Thus, we now present the white matter connections of each of the right hemisphere fROIs of both networks in new **Supplementary Figs. 10, 12, 17, 21**. Results are largely consistent across hemispheres.

It is unclear why the authors segmented the SLF and AF in 30 nodes. Why 30? They should justify why they used this type of segmentation.

Thank you for your comment. The purpose of the segmentation of the SLF and AF to nodes is to analyze tract properties along the tract, rather than providing one single value that summarizes the tract. Previous research has typically segmented fascicles into 100 nodes (e.g. Teubner-Rhodes et al. J Neurosci, 2016; Yeatman et al. NComms, 2018). We were concerned that, with such a fine-grained segmentation, the decoding results in adjacent nodes presented in Fig. 4c,f,i may not be independent. That is, if the tracts are sampled too finely, the training and testing nodes may originate from the same voxel. In order to address your concern, we now segment the fascicles into 100 nodes for all analyses (see Fig. 4-5), similar to previous studies. For the decoding analysis, we use the fifth more anterior node, rather than the directly neighboring node, to ensure independence of training and test data. Results remained the same regardless of the number of nodes used.

The paper draws suggests numerous implications that are too overstated given the current data. Firstly, the authors indicate that their paper reveals something about comorbidity of learning disorders. This is not at all the case: they collected their data in typically developing individuals and the participants under study are adults (who have developed these abilities) but not children. As a result, this paper cannot really tell something about development or about comorbidity. Second, the authors claim that their findings provide evidence for the triple-code model. This is not the case. For many reasons. One is that the triple-code model is about how we process numbers in different codes (verbal, visual, analogue) but this is not what the authors investigated. Instead they focused on calculation, and not on number processing per se. It is true that different types of calculation, i.e. different operations, could be linked indirectly to the different codes. But as the authors only examined one type of operation (which may be the closest to the verbal code), their data cannot speak to this issue.

Thank you for your comment. In order to address your concern, we have extensively edited the manuscript. First, we accept your comment on the triple-code theory and removed the triple-code model from the manuscript. Second, we removed claims that our data reveals something about comorbidity of learning disabilities from the manuscript. Instead, we now suggest that future clinical studies could test how our results relate to math and reading learning disabilities.

In the discussion, on page 26, we now write: *“This last finding makes an interesting prediction for potential links between white matter properties and math and reading skills: We hypothesize that the properties of the inferior and superior sections of the SLF and AF may independently contribute to reading and math performance. That is, if myelination improves transmission of information across distributed networks, then T_1 of the inferior portion of the SLF and AF may correlate with reading ability, while T_1 of the superior portions of these fascicles may correlate with math ability. Accordingly, atypical myelination of these tracts within the superior and inferior portion of the SLF and AF during development may also be associated with math or reading learning disabilities, respectively. Future studies with clinical populations that simultaneously evaluate the neural substrates of both math and reading learning disabilities can test these hypotheses.”*

The authors highlight that segmenting large white matter tracts into sub-bundles is important to further our theoretical understanding of how networks are supported by white matter tracts. I agree, but they should clarify what these segments can tell us about these networks. How should we interpret them?

Thank you for your comment. We now discuss what these sub-bundles can tell us about these networks in the discussion, where we write on page 25: *“Our study yields novel insights on the fascicles of the reading and math networks. First, we show that the SLF and AF are shared across the math and reading*

networks. Second, even as the SLF and AF are key fascicles for both tasks, they each contain separate sub-bundles for reading or math. Specifically, analogous to separate lanes on a highway, parallel and segregated tracts within these fascicles are part of either the reading or the math network (Fig. 4, Supplementary Fig. 14-16). These distinct sub-bundles for math and reading within the SLF and AF suggest that the white matter connections of the math and reading networks are more spatially specific than was previously believed."

MINOR ISSUES

Page 3, the authors suggest reasons for overlap between reading and math. One area of overlap, which can be explicitly linked to the arcuate fasciculus as well as activity in temporoparietal cortex is the hypothesis that arithmetic facts are phonologically represented, and that this might represent an overlap between reading and arithmetic and might explain why reading and arithmetic correlate and why their disorders co-occur. Many studies have investigated this (including neuroimaging work) and I feel this work should be credited.

Thank you. We now refer to the suggested literature briefly in the introduction, on page 3, and more extensively in the discussion, on pages 24-25.

In the introduction we now write: *"While math and reading are distinct tasks, they utilize several overlapping cognitive processes, including encoding of visual stimuli, verbalization, as well as working memory¹. It has been suggested that the degree to which brain activations related to math and reading overlap may depend on the arithmetic task. For example, brain responses related to arithmetic fact retrieval (e.g. during addition of small numbers) overlap more with brain responses related to reading than responses related to procedural-based computations²."*

In the discussion we now write: *"It should be noted though, that the fascicles of the math and reading networks reported here likely do not contain the entire white matter of these networks, for three reasons. [...] Third, here we focused on addition, and did not investigate neural substrates of other arithmetic operations (subtraction, multiplication, and division) or broader mathematical abilities. Previous work has shown that different arithmetic operations vary in their neural substrates^{25,53,54} and that addition of small numbers relies more heavily on arithmetic fact retrieval compared to other arithmetic operations such as subtraction⁵⁵. Arithmetic fact retrieval, in turn, has been linked to phonological processing^{25,56,57}, which is also involved in reading (for review see^{6,58}). These findings are in line with the observation that, compared to other arithmetic tasks, brain activations induced by addition show more overlap with brain activations induced by reading². Here, we chose to investigate addition, using number-letter morphs, in order to match the mathematical and reading tasks as much as possible. Future studies that examine the gray and white matter substrates of other arithmetic operations and broader mathematical abilities will shed important light on which components of the revealed math network are specific to addition and which components extend beyond this arithmetic task."*

The authors indicate that their reading and math task are matched. I do not fully agree, since the math task is only single-digit addition and is, to a lesser extent, reflective of mathematical processing per se. On a related note, the authors focus in their work on arithmetic and not mathematics, the latter is a much broader ability.

Thank you for your comment. In order to address this concern, we now include behavioral data, showing participants' performance and response times in the various tasks.

In the first paragraph of the results section, on page 6-7, we now write: *"Participants successfully performed all tasks in the experiment (average accuracy (\pm SE): 88.16(2.43)%). Both accuracy and response*

*times (RTs) differed across the reading, adding, and color tasks (main effect of task: accuracy: $F(2,38)=10.30$, $p=0.0003$, $\eta^2=0.35$; RTs: $F(2,38)=72.20$, $p<0.0001$, $\eta^2=0.79$). While accuracy was significantly higher in the reading task, relative to the other two tasks (all $ps<0.002$ after Bonferroni correction, n.s. between adding and color), response times were shortest in the adding, intermediate in the reading task, and slowest in the color task (all $ps<0.003$ after Bonferroni correction). It is unlikely that performance differences across tasks drove responses across cortex for two reasons: i) within a task, response accuracy and neural task preference, i.e. the extent of preferential activations for a given task, did not show any clear relationship across participants (parameter maps presented in **Supplementary Fig. 1-4** are sorted according to task performance, for group maps see **Supplementary Fig. 5**), and ii) accuracy and response times were not consistently different across tasks, yet we could identify task-selective functional regions of interest (fROIs) for all tasks (**Fig. 1b**, **Supplementary Fig. 6-7**)."*

We now also highlight in the discussion that the paper focuses on arithmetic, where we write on page 24: *"Third, here we focused on addition, and did not investigate neural substrates of other arithmetic operations (subtraction, multiplication, and division) or broader mathematical abilities."*

Were all subjects right-handed? Please clarify.

We now added handedness information to the methods section where we write, on page 27: *"20 volunteers (10 female, mean age \pm SE: 27 ± 1 years, 1 left-handed) were recruited from Stanford University and surrounding areas and participated in two experimental sessions."*

Please provide more details on the motion assessment of the fMRI data. Add the criteria and report how many there were discarded.

We now added information on the motion assessment of the MRI data to the method section, on page 27 we now write: *"No subject exceeded our threshold of 3 voxels within-run motion and 3.5 voxels between run motion in the fMRI and dMRI sessions; for the qMRI session we visually inspected the quality of the resulting output maps and did not exclude any subjects due to bad data quality."*

Reviewer #2 (Remarks to the Author):

The current manuscript attempts to map the white matter fascicles involved in reading and mathematical processing. The authors used a task that they very recently developed to identify cortical fROIs associated with reading and math, and used AFQ to identify 12 major fascicles. They next dilated the tract endpoints by 7mm, and the resulting tracts that intersected with any fROI were designated to "functionally-defined white matter tracts." The authors report the fascicles that each domain is associated with, and that the individual tractography streamlines are segregated within these fascicles.

My largest concern is that the approach used here relies on an enormous leap of faith in the assumption that (1) a region's proximity to a fascicle endpoint indicates that its efferent or afferent axons travel through that nearby fascicle, and (2) that tracts only enter and exit a fascicle at its endpoints. The authors would need to provide a substantial degree of prior anatomical evidence to show that all gray matter within 7mm of a fascicle terminus are connected to it, and that axons always enter or exit a fascicle at its terminus. Otherwise, the authors need to directly seed the individual gray matter fROIs for tractography, which a very large corpus of prior work has done. This will be extremely important in order to know what regions each fROI is connected to, as well as which fascicles are used to connect them.

Thank you for your comment. You raise the concern that our original methods may not have been accurate enough to identify tracts that are specific to each fROI as we included all gray matter within 7mm of the fiber tract endpoint. To address your concern, we have changed our dMRI analysis approach substantially and re-analyzed all dMRI data from scratch. This allowed us to fully remove any dilation of fiber tract endpoints, thereby increasing the spatial accuracy of our measurements.

In the new approach we implemented several changes: (1) We upgraded our software to MRtrix3, which also allowed us to use anatomical constrained tractography (ACT, Smith et al., Neuroimage, 2012). ACT utilizes information of different tissue types from the FreeSurfer (<https://surfer.nmr.mgh.harvard.edu/>) segmentation of each participant's high-resolution anatomical scan to optimize tractography. (2) ACT also allowed us to identify the gray-white matter interface (GWMI) directly underneath the fROIs, and we seeded streamlines in this interface, rather than in the entire white matter mask. This enabled us to focus on those fiber tracts that reach the gray matter from the get-go. (3) In order to determine the fWMTs of each fROI, we identify fiber tracts (i.e. streamlines) that are seeded or terminate in the GMWI underneath each fROI, without dilating the tract endpoints (**Fig. 1c**). Tractography is not reliable in the gray matter, so it is recommended to perform the intersection between tracts and fROIs in the white matter just below the functional regions of interest (see Bouhali et al (2014); Lerma-Usabiaga et al (2018)). (4) To identify tracts that support pairwise connections between fROIs (**Fig. 3-5**), we intersected the fWMTs of each fROI with the GWMI underneath the other fROI, again without dilating tract endpoints. (5) We no longer use LiFE (Pestilli et al., 2014), as LiFE necessitates a whole-brain connectome with seeding in the entire white matter, in order to build a model of the diffusion data, which we no longer generate. Instead, we optimize our connectomes with AFQ, by removing fiber tracts that are further than 4 SD away from the core of their respective fascicle, similar to Yeatman et al. 2012. Critically, this new analysis pipeline does not dilate the fiber tracts at any point.

In general, despite the change in methodological approach, the new fWMT travel through the same fascicles we identified in the original manuscript. The biggest changes that resulted from the new approach were (i) an increase in fWMTs associated with the pAF and the ILF, and (ii) a decrease in fWMTs that were associated with the VOF (**Fig. 2**). Critically, our main results showing segregation of math- and reading-related tracts as well as structural differences between these sub-bundles within fascicles remain unchanged (**Fig. 3-5**).

Additionally, we implemented two new control analyses to support the robustness of our results. First, to ensure that results are not affected by variations in size of fROIs across regions and participants, we performed a control analysis using the methodological approach described above with constant-size spherical ROIs (radius of 7mm, this radius was chosen based on previous studies, e.g. Yeatman et al. 2013) centered on the fROIs (results are presented in **Supplementary Fig. 11, 12, 18, 22**). Second, following your suggestion, we also implemented direct fROI-to-fROI tractography for those analyses that focus on specific pairwise connections (i.e. those associated with **Fig. 4-5**). Here, we generated tracts between pairs of fROI where one fROI was used as a seed and the other fROI was used as a target for tractography. The resulting pairwise tracts were then classified with AFQ, as in our main approach (results are presented in **Supplementary Fig. 19, 23**). We report these new methods on page 38 of the methods section. Results of these control analyses are similar to the data presented in the main manuscript.

Other methodological concerns:

You report that you acquired data from 14 subjects, but most figures and text refer to 12 subjects. Why were 2 subjects excluded?

Thank you for following up on this. We did not exclude any subjects. However, we could not always identify all fROIs in all participants. When an fROI could not be identified, the respective white

matter connections could not be evaluated for that particular subject. For this reason, not all subjects contributed to all analyses. Please also note that the number of subjects has now changed as we scanned additional subjects. Presently the number of subjects is 20.

Please also include subject handedness.

We now added handedness information to the methods section where we write, on page 27: *“20 volunteers (10 female, mean age \pm SE: 27 \pm 1 years, 1 left-handed) were recruited from Stanford University and surrounding areas and participated in two experimental sessions.”*

At the end of pg 24: *“we report data from regions that showed a reliable preference across participants”* and *“here we focus on the most robust activations.”* Please provide an operational definition that delineates exactly how you chose these 8 fROIs.

Thank you for your comment. We now provide an operational definition in the methods section, where we write on page 30: *“For all analyses, we report data from regions that showed a reliable preference for a given task across subjects. That is, we report regions that could be identified in the left hemisphere in at least 90% of the participants. In other words, while in a given individual there may be additional voxels that respond preferentially during reading and/or during math, here we focus on the most consistent activations.”*

Also on pg 24: *“regions were defined in each participant’s cortical surface using both functional and anatomical criteria.”* What were those anatomical criteria?

Thank you for your clarifying question. By anatomical criteria we meant that we use anatomical landmarks as guideline. We now provide a clarifying example of this approach in the methods section, where we write on page 30: *“Reading- and math-related gray matter regions of interest (fROIs) were defined in each participant’s cortical surface using both functional and anatomical criteria. For example, for our IFG fROI we took only those voxels that (i) showed the relevant task preference beyond the threshold of $T \geq 3$ and (ii) fell within the inferior frontal gyrus.”*

Please list the 12 fascicles that you initially identified with AFQ. It would also be helpful to include a supplementary figure that illustrates each of them.

Thank you for your suggestion. We realized that AFQ also separates the Forceps into Forceps Major and Forceps Minor, so we actually extracted 13 fascicles with AFQ. We now corrected this in the manuscript. To address your comment, we also generated a new **Supplementary Fig. 9** showing the fascicles identified by AFQ in both hemispheres of a representative subject.

For the svm analysis: why did you use a 2nd degree polynomial instead of linear? What were your other hyperparameters? How were they chosen/optimized?

Thank you for your clarifying question. This choice was made because the 2nd degree polynomial slightly outperformed the linear classifier, but the performance difference is negligible. We have revised the SVM analysis, which now uses the default linear SVM classifier implemented in Matlab instead.

New **Fig. 4** shows the results of analyses using the new approach to identify fWMTs and implementing the linear SVM. Please note that the choice of classifier affected the numerical values but did not have a significant impact on the obtained results or the conclusions.

Since your analyses are based on fROI-fascicle endpoint distance, they will be sensitive to inter-ROI distance. What are the distance minima between each neighboring fROI (ie IFG/PCS, STS/ITG/OTS, SMG ROIs) within the volume?

Thank you for your comment. Indeed, inter-ROI distance might influence the pairwise connectivity, generating a bias for neighboring ROIs. For this reason, we never evaluate the pairwise connections between ROIs that are neighboring (e.g., between IFG and PCS). Our analyses are largely geared to examine long-range connections, through fascicles, such as connections between lobes. For example, the SLF connects the parietal and frontal lobes. For this reason, we acknowledge on page 24 of the discussion that we do not provide a comprehensive connectivity map of all white matter fibers of either the math or reading networks.

You do not specify what MR-spaces each analysis was performed in. Was dMRI data transformed to anatomical? Was tractography performed in native dMRI space? Were fROIs transformed to anatomical? How did you perform all of these registrations and transformations?

Thank you for following up on this. All analyses were performed in the native brain space of each individual and each data type. Data of all types (fMRI, dMRI, qMRI) were aligned to the 1 mm whole brain anatomical volume of each individual. We now clarify the details in the methods section.

On page 28 we write: *“Each participant’s anatomical brain volume was used as the common reference space for all analyses, which were always performed in individual native space.”*

On page 28 we write: *“The functional data was analyzed using the mrVista toolbox (<http://github.com/vistalab>) for Matlab, as in previous work³⁶. In short, the data was motion-corrected within and between scans and then manually aligned to the anatomical volume. The manual alignment was optimized using robust multiresolution alignment⁶⁹.”*

On page 31 we write: *“DMRI data was preprocessed using a combination of tools from mrTrix3 (github.com/MRtrix3/mrtrix3) and mrDiffusion toolbox (<http://github.com/vistalab>) for Matlab. [...] Third, dMRI data was registered to the average of the non-diffusion weighted images and aligned to the corresponding high-resolution anatomical brain volume using rigid body transformation.”*

On page 37 we write: *“Quantitative MRI (qMRI⁴¹) data was collected within the same session and with the same head coil as the dMRI data. T₁ relaxation times were measured from four spoiled gradient echo images with flip angles of 4°, 10°, 20° and 30° (TR: 14 ms, TE: 2.4 ms). The resolution of these images was later resampled from 0.8x0.8x1.0 mm³ to 1mm isotropic voxels, and qMRI data was aligned with the high-resolution anatomical scan using rigid body transformation.”*

Results:

Please also include accuracy & RT for each subject, as well as whole brain contrast maps (as heatmaps as opposed to binary labels) for each subject, matched in order as the aforementioned behavioral data

Thank you. We now include behavioral data in the first paragraph of the results section, where we write on pages 6-7: *“Participants successfully performed all tasks in the experiment (average accuracy (±SE): 88.16(2.43)%). Both accuracy and response times (RTs) differed across the reading, adding, and color tasks (main effect of task: accuracy: $F(2,38)=10.30$, $p=0.0003$, $\eta^2=0.35$; RTs: $F(2,38)=72.20$, $p<0.0001$, $\eta^2=0.79$). While accuracy was significantly higher in the reading task, relative to the other two tasks (all $ps<0.002$ after Bonferroni correction, *n.s.* between adding and color), response times were shortest in the adding, intermediate in the reading task, and slowest in the color task (all $ps<0.003$ after Bonferroni correction). It is unlikely that performance differences across tasks drove responses across cortex for two reasons: i) within a task, response accuracy and neural task preference, i.e. the extent of*

*preferential activations for a given task, did not show any clear relationship across participants (parameter maps presented in **Supplementary Fig. 1-4** are sorted according to task performance, for group maps see **Supplementary Fig. 5**), and ii) accuracy and response times were not consistently different across tasks, yet we could identify task-selective functional regions of interest (fROIs) for all tasks (**Fig. 1b**, **Supplementary Fig. 6-7**)."*

This paragraph also refers to the requested heatmaps, which can be found in **Fig. S1-S4**.

The fROIs appear to be extremely variable in size across regions and subjects. fROI size will unquestionably affect the likelihood that a dilated fascicle endpoint will intersect that fROI. If you are using anatomical constraints, it would be best to use a percentile-based threshold (e.g. top 10%) rather than an absolute threshold ($t=3$), which will guarantee similar sized fROIs across subjects (and fROIs if the anatomical constraints are the same size).

Thank you for this suggestion. To address your concern, we repeated all analyses with constant-size spherical ROIs with a radius of 7 mm. A radius of 7 mm was chosen as it was used in previous studies (e.g. Yeatman et al 2013; Klein et al 2013). Results with spherical ROIs are shown in new **Supplementary Fig. 11, 12, 18, and 22**. Our main results and conclusions remain the same when using constant size fROIs across regions and subjects.

Please provide a group-average heatmap of math and reading responses, which will help the reader evaluate the reliability of these regions across subjects.

Thank you for your suggestion. We now present group-average heatmaps in new **Supplementary Figure 5**.

For the analyses in Fig3: this would be more clearly interpretable in the context of between vs within tasks, as opposed to simply within task. Please repeat this analysis for math->reading fROIs as well. In any case, since the question here is whether math/reading fROIs preferentially connect to other math/reading fROIs, and not whether they are both 7mm away from coinciding fascicular endpoints, this would be much better if each region was a seed and target for each other region. Currently, high DC just means that both regions are near the endpoints of fascicle, and not that they are connected to each other.

Thank you for suggesting this complementary analysis. We now present both within and between network connectivity in **Fig. 3**. On average, the dice coefficient (DC) for within network connectivity significantly exceeded the DC for between network connectivity. We report these results on pages 15 and 17, where we write: "*Analyzing pairwise connections across fROIs of the math and reading networks revealed significantly above chance DC (**Fig. 3n**, Bonferroni adjusted threshold of $p<0.004$) between: (i) the OTS and the IPS (paired t-test: $p<0.0001$, $t(17)=5.10$), (ii) the OTS and SMGm (paired t-test: $p=0.0007$, $t(17)=4.14$), (iii) the OTS and the PCS (paired t-test: $p=0.003$, $t(15)=3.59$), (iv) the STS and the PCS (paired t-test: $p=0.001$, $t(17)=3.96$), (v) SMGr and the PCS (paired t-test: $p=0.0002$, $t(17)=4.72$), (vi) the IFG and the ITG (paired t-test: $p<0.0001$, $t(19)=5.40$), and (vii) the IFG and SMGm (paired t-test: $p<0.0001$, $t(19)=5.06$). Similar to the within-network connections described above, the pAF supported temporal-parietal between-network connections (OTS-IPS and OTS-SMGm), the SLF supported frontal-parietal between-network connections (SMGr-PCS and SMGm-IFG), and the AF supported frontal-temporal between-network connections (OTS-PCS, ITG-IFG) (data not shown). These connections are summarized in a schematic (**Fig 3o**). [...] Nonetheless, within-network pairwise connectivity (as quantified with the DC) was significantly higher than between-network connectivity (paired t-test comparing average within-network and between-network DCs: $p=0.01$, $t(15)=2.83$)."*

Please also note that in our new dMRI analysis approach there is no dilation of tract-endpoints and that we intersected individual tracts (i.e. streamlines), not entire fascicles with fROIs at the GWMI. AFQ was used to determine which fascicle each long-range tract belongs to.

In additional control analysis, we also implemented the approach you suggested and used one fROI as a seed and another fROI as a target for tractography in order to determine their pairwise connections. This was done for the analyses presented in **Fig. 4 and Fig. 5**. For the analyses in **Fig. 3** this approach was not feasible because, in order to determine the relative connectivity weights of each pairwise connection, we needed to start with a single connectome that includes all connections.

For the analyses in Fig4, the simplest thing to do would be to identify all voxels in each tract group, and compute an overlap metric such as DICE over the sets of voxels occupied by each group of tracts. Then test against the null hypothesis that they overlap completely ($DC = 1$) or against an empirical null distribution via permutation, resubstitution, or something similar. The metrics used currently are indeed interesting but rather complicated, and a more parsimonious test of overlap would be preferred.

Thank you. Please note that here we are not testing if tracts from the math and reading network are overlapping (i.e. if there are any tracts which connect all four fROIs) but rather if tracts from the math and reading networks are spatially segregated or intertwined. This question cannot be answered with the DICE coefficient, which is a measure of the overlap between two samples. This is the reason why we chose the two complimentary approaches presented in **Fig. 4**. Moreover, a DC analysis could also only be performed by identifying the voxels each tract group travels through, which would significantly reduce the resolution of the resulting measurement, potential causing artificial overlap between the sub-bundles.

Lastly, these subjects are undeniably identical with those from the same authors' 2018 NeuroImage paper, both in response patterns, and in cortical folding and surface topography. In that paper, there were 15 subjects. Why was one subject excluded?

Thank you for your comment. There is overlap among the subjects in both studies, but they are not identical. After completing our 2018 NeuroImage study, we invited all participants from that study to participate in a new scanning session in which we collected dMRI and qMRI data for the current study. 12 of the original 15 subjects participated in this additional scanning session, the remaining 3 subjects did not. One subject had moved away, the other 2 had simply lost interest. Additionally, in response to a comment by reviewer 1, we have now scanned 6 additional subjects for the revision. Accordingly, there are now 8 new subjects that were not included in the 2018 NeuroImage paper. We now clarify this in the methods section, where we write on pages 27-28: *"20 volunteers (10 female, mean age \pm SE: 27 \pm 1 years, 1 left-handed) were recruited from Stanford University and surrounding areas and participated in two experimental sessions. [...]. A subset (N=12) of the fMRI data were also used for our previous study.³⁶"*

I greatly hope that the authors do not simply resubmit this manuscript elsewhere as-is, and take time to re-evaluate the analyses that they employed to answer the questions they are attempting to address.

Thank you for your feedback. We have taken the time to extensively re-evaluate and revise our analyses, which we believe strengthen our results and make them more precise. We are hopeful that you will find this new version of the manuscript suitable for publication.

Reviewer #3 (Remarks to the Author):

The authors report on an innovative study using a multimodal approach involving first functional MRI for locating activity related to math and reading, followed by diffusion MRI and quantitative MRI to define their connected fascicles. The study is technically strong, the findings are interesting, and the manuscript is suitable to a wide readership as it presents a major advance in understanding subdivisions of fascicles. There are several areas where clarification is needed.

Main Concerns:

The authors refer to arithmetic, yet the task involves only addition. It has been proposed that addition (and multiplication) of small numbers is more similar to reading in terms of overlap (due to verbal retrieval mechanisms used for both) and that subtraction (and division) is different from these, relying more on procedural-based computation. It seems therefore that the manuscript needs to mention that the findings are specific to addition and not arithmetic at large.

Thank you for your comment. In order to address this concern, we now highlight the fact that our results are specific to addition, on page 3, i.e. in the introduction, and more extensively on pages 24-25, i.e. in the discussion.

In the introduction we now write: *“While math and reading are distinct tasks, they utilize several overlapping cognitive processes, including encoding of visual stimuli, verbalization, as well as working memory¹. It has been suggested that the degree to which brain activations related to math and reading overlap may depend on the arithmetic task. For example, brain responses related to arithmetic fact retrieval (e.g. during addition of small numbers) overlap more with brain responses related to reading than responses related to procedural-based computations².”*

In the discussion we now write: *“It should be noted though, that the fascicles of the math and reading networks reported here likely do not contain the entire white matter of these networks, for three reasons. [...] Third, here we focused on addition, and did not investigate neural substrates of other arithmetic operations (subtraction, multiplication, and division) or broader mathematical abilities. Previous work has shown that different arithmetic operations vary in their neural substrates^{25,53,54} and that addition of small numbers relies more heavily on arithmetic fact retrieval compared to other arithmetic operations such as subtraction⁵⁵. Arithmetic fact retrieval, in turn, has been linked to phonological processing^{25,56,57}, which is also involved in reading (for review see^{6,58}). These findings are in line with the observation that, compared to other arithmetic tasks, brain activations induced by addition show more overlap with brain activations induced by reading². Here, we chose to investigate addition, using number-letter morphs, in order to match the mathematical and reading tasks as much as possible. Future studies that examine the gray and white matter substrates of other arithmetic operations and broader mathematical abilities will shed important light on which components of the revealed math network are specific to addition and which components extend beyond this arithmetic task.”*

Given the overlap between reading and addition, it seems strange that the investigation focused on read>math and vice versa. What about areas where function is reported for both? It would be critical to examine the fibers connected to regions that show functional overlap (conjunction) for math and reading and test if they are segregated or intertwined. Without this information we still do not have the full picture of how these two skills are shared versus separate. By only looking at areas that are functionally segregated, there seems to be a greater likelihood that they are also anatomically segregated.

Thank you for suggesting this additional analysis. We liked this idea and added the conjunction analysis to our manuscript (see **Figs 2-5**). The conjunction analysis revealed several regions in the brain (in the IPS, SMG, PCS, and IOTC) that are more strongly activated during math and reading tasks than in the color task (**Supplementary Fig. 6-7**). Interestingly, the regions detected in the conjunction analysis substantially overlap with the math network. In fact, a direct comparison of responses to math and reading in these regions shows that while they are activated in both tasks, they are activated more strongly during the math than the reading task (**Supplementary Fig. 8**). The only exception was a cluster in the lateral occipito-temporal cortex (IOTC), which showed no preference for either task. Indeed, the IOTC overlapped with both a region in the ITG which is involved in math as well as a region in the OTS which is involved in reading (**Supplementary Fig. 6-7**). Given that the other fROIs overlap with the math network, to examine additional connections that were not included in our prior analyses and may be involved in common processing to both math and reading, we evaluated the white matter connections of the IOTC fROI in subsequent analyses, results shown in **Fig. 2-5**. Critically, even though the IOTC is involved in both math and reading, tracts within the AF that connect the IOTC to the PCS in the math network remain segregated from those that connect the IOTC to the IFG in the reading network (**Fig. 4**). This strengthens our conclusion about parallel processing streams for math and reading.

In the results section we now write:

On page 7: *“We also considered regions involved in both math and reading, defined by the conjunction of both higher responses during math than color as well as during reading than color task (conjunction analysis, $\text{math} > \text{color} \cap \text{reading} > \text{color}$).”*

On page 10: *“Four regions in the brain showed higher responses during both the adding and reading tasks compared to the color task (conjunction analysis, $\text{math} > \text{color} \cap \text{reading} > \text{color}$; **Fig. 1b-orange, Supplementary Fig. 6-7**): (i) A region in the intra-parietal sulcus (IPS; left hemisphere: $N=18$, $\text{size} \pm \text{SE}$: $363 \pm 90 \text{ mm}^3$; right hemisphere: $N=16$, $\text{size} \pm \text{SE}$: $251 \pm 63 \text{ mm}^3$). (ii) A region in the SMG (SMG, left hemisphere: $N=20$, $\text{size} \pm \text{SE}$: $434 \pm 89 \text{ mm}^3$; right hemisphere: $N=17$, $\text{size} \pm \text{SE}$: $249 \pm 54 \text{ mm}^3$). (iii) A region in the inferior part of the precentral sulcus (PCS; left hemisphere: $N=19$, $\text{size} \pm \text{SE}$: $350 \pm 87 \text{ mm}^3$; right hemisphere: $N=18$, $\text{size} \pm \text{SE}$: $111 \pm 26 \text{ mm}^3$). (iv) A region in the lateral occipito-temporal cortex (IOTC) that extended from the ITG to the OTS (left hemisphere: $N=19$, $\text{size} \pm \text{SE}$: $720 \pm 141 \text{ mm}^3$; right hemisphere: $N=18$, $\text{size} \pm \text{SE}$: $241 \pm 53 \text{ mm}^3$). Except for the IOTC region, these fROIs were small and tended to overlap with the math fROIs (**Fig 1b, Supplementary Fig. 6-7**). Indeed, responses in the IPS, PCS, and SMG conjunction fROIs were significantly stronger during the adding task than the reading task (**Supplementary Fig. 8**; ANOVA with hemisphere, task and stimulus as factors; main effect of task: IPS: $F(1,14)=17.30$, $p=0.001$, $\eta^2=0.55$; PCS: $F(1,16)=12.97$, $p=0.002$, $\eta^2=0.45$; SMG: $F(1,16)=19.37$, $p=0.0004$, $\eta^2=0.55$). The only conjunction fROI in which responses during adding and reading did not differ significantly was the IOTC (main effect of task: $F(1,17)=3.83$, $p=0.07$, $\eta^2=0.18$). This region overlaps with regions in both the math network (in the ITG) and the reading network (in the OTS). Thus, in subsequent analyses we considered the IPS, PCS, and SMG regions as part of the math network, but the IOTC as a conjunction region involved in both tasks.”*

The manuscript does not report on the performance (accuracy and reaction time) of the fMRI task. Are the three tasks matched on performance? One concern is that the reading task is hard (it is difficult to read the “letters”). Another is that the memory load on the reading and math tasks are different: the reading task relies more on short term memory while the math task relies on working memory because of the manipulation performed during addition.

Thank you for your comment. We now report on accuracy and reaction time data in the first paragraph of the results section, where we write on pages 6-7: *“Participants successfully performed all tasks in the experiment (average accuracy ($\pm \text{SE}$): $88.16(2.43)\%$). Both accuracy and response times (RTs) differed across the reading, adding, and color tasks (main effect of task: accuracy: $F(2,38)=10.30$, $p=0.0003$, $\eta^2=0.35$; RTs: $F(2,38)=72.20$, $p<0.0001$, $\eta^2=0.79$). While accuracy was significantly higher in the reading task, relative to the other two tasks (all $ps<0.002$ after Bonferroni correction, n.s. between adding and color), response times were shortest in the adding, intermediate in the reading task, and*

*slowest in the color task (all $p < 0.003$ after Bonferroni correction). It is unlikely that performance differences across tasks drove responses across cortex for two reasons: i) within a task, response accuracy and neural task preference, i.e. the extent of preferential activations for a given task, did not show any clear relationship across participants (parameter maps presented in **Supplementary Fig. 1-4** are sorted according to task performance, for group maps see **Supplementary Fig. 5**), and ii) accuracy and response times were not consistently different across tasks, yet we could identify task-selective functional regions of interest (fROIs) for all tasks (**Fig. 1b**, **Supplementary Fig. 6-7**)."*

Minor Concerns:

Abstract. The abstract is very broad and would benefit from removal of the broader claims at the beginning and the end and instead provide more details on the details of the study.

Thank you for your feedback. We have extensively edited the abstract; we removed some of the broader claims and instead fleshed out our results. The abstract now reads as follows: *"Math and reading involve distributed brain networks and have both shared (e.g. encoding of visual stimuli) and dissociated (e.g. quantity processing) cognitive components. To date, it is unknown what are shared vs. dissociated gray and white matter substrates of the math and reading networks. Here we address this question using an innovative, multimodal approach applying functional MRI, diffusion MRI, and quantitative MRI to define these networks and evaluate the structural properties of their fascicles. Results reveal that i) there are distinct gray matter regions which are preferentially engaged in either math or reading and ii) the superior longitudinal (SLF) and arcuate (AF) fascicles are shared across math and reading networks. Strikingly, within these fascicles, reading- and math-related tracts are segregated into parallel sub-bundles and show structural differences related to myelination. These novel findings open a new avenue of research that examines the contribution of sub-bundles within fascicles to specific behaviors."*

Introduction. "The VWFA is thought to process visually presented words^{18,19} and is causally involved in word recognition, as lesioning it produces dyslexia²⁰ " Better to use the term acquired dyslexia or alexia as dyslexia usually refers to developmental dyslexia.

Thank you, we corrected this.

Para beginning with "To date, it is unknown if these fascicles, which are important for reading, are also associated with mathematical processing." This needs to be preceded by information on how reading and math are connected. Without it, it may seem odd to the reader why the same fascicles would be associated with reading and math.

Thank you, we have changed this sentence. It now reads as follows: *"Although it has been suggested that math and reading utilize several overlapping cognitive processes¹, presently it is unclear if fascicles associated with reading are also associated with mathematical processing."*

"Third, to our knowledge, no study has examined the relationship between gray and white matter substrates of the math network. That is, it is unknown which fascicles intersect with cortical regions involved in mathematical processing." this statement does not seem to fit with the two goals that follow – as they do not describe cortical regions, just white matter. Please also note prior work referenced below.

Thank you for your comment. We rephrased this entire paragraph to ensure that (i) it properly credits prior work and (ii) so that the goals align with the rest of the paragraph. We now write: *"Although it has been suggested that math and reading utilize several overlapping cognitive processes¹, presently it*

is unclear if fascicles associated with reading are also associated with mathematical processing. This fundamental gap in knowledge is due to four main reasons: First, substantially more neuroscience research has been done on the neural bases of reading^{4,5} than the neural bases of math⁶. Second, most prior studies have evaluated either the neural bases of math^{7,24–27} or the neural bases reading^{18,21,28–33}, but not both systems within the same individuals (for an exception see²). Third, no study directly examined the white matter connections associated with functional regions involved in mathematical processing within the same individuals. Previous work on the neural substrates of mathematical processing has focused on linking gray matter structure, evaluated with voxel-based morphometry, and white matter (³⁴, for review see^{6,7}). The only study that evaluated the white matter connections of functional regions related to mathematical processing evaluated gray and white matter in distinct groups of participants³⁵. While this study provided first insight on the white matter connections of the math network, more work is needed to understand these connections within the same individuals. In contrast, a few previous studies have examined which white matter fascicles connect to a key cortical region of the reading network, namely the VWFA, evaluating functional regions and white matter connections within the same subjects^{18–20}. Thus, the goal of this study is twofold: (1) identify and quantify the white matter connections of cortical regions involved in mathematical processing within individual subjects and (2) determine which aspects of this white matter are unique to the math network, and which are shared with the reading network.”

Methods. Were the scans conducted on the same day?

Thank you for your comment. The dMRI and qMRI data were collected in the same session, the fMRI data was collected in a different session and on a different day. We now clarified this in the methods section, where we write on pages 31 and 37: “Diffusion-weighted MRI (dMRI) data was collected in the same participants during a different day than the fMRI data, at the same facility and with the same 32-channel head-coil. [...] Quantitative MRI (qMRI⁴¹) data was collected within the same session and with the same head coil as the dMRI data.”

If I understand it correctly, the scans did not cover the entire cortex- which areas were omitted?

Generally, our scans covered the entire cortex. In cases where the participants head was unusually large, it is possible that the very superior part of the cortex, specifically regions related to motion, may have been truncated.

Discussion. The discussion makes many different points using numerical text features. This feels unnecessary. For example, the authors should identify the main findings (two to three) and then turn to a broader discussion which includes other information, without trying to highlight each as a major achievement. Similarly, the fact that the results align with expectations set up by the triple code is highlighted as an important finding, but it seems that the results of a study on arithmetic should align with the most prominent model of math processing. It would be awkward if they did not, so this observation could be toned down. Likewise, there seems to be a strong claim of novelty all around which is not necessary since the aspects that are novel are clear; overdoing it may give the appearance that some prior work has been overlooked. For example, the authors should probably acknowledge the following study using functional seed regions for fiber tracking:

Klein, E., Moeller, K., Glauche, V., Weiller, C., & Willmes, K. (2013). Processing pathways in mental arithmetic-evidence from probabilistic fiber tracking. *PLoS One*, 8(1), e55455.

<https://doi.org/10.1371/journal.pone.0055455>

Thank you for your feedback. In order to address this concern, we have toned down our statements regarding the novelty of the current data and the conclusion that can be drawn from it. For

instance, we completely removed the paragraph linking our results to the triple-code model. Further, in accordance with feedback from reviewer 1, we removed all claims relating our data to comorbidity of math and reading learning disabilities, as we are only testing typically performing adults in the current study. We retained some of the numerical text features you highlighted because we feel that for researchers outside of the field the novel aspects of the current study may not be as immediately apparent and for those readers a quick numerical overview may be helpful. We do dedicate a larger portion of the discussion to previous research, though. Additionally, we now refer to the contribution of Klein et al. (2013), which we had previously overlooked, in both the introduction and the discussion.

In the Introduction we now write on page 4: *“Previous work on the neural substrates of mathematical processing has focused on linking gray matter structure, evaluated with voxel-based morphometry, and white matter (34, for review see6,7). The only study that evaluated the white matter connections of functional regions related to mathematical processing evaluated gray and white matter in distinct groups of participants35. While this study provided first insight on the white matter connections of the math network, more work is needed to understand these connections within the same individuals.”*

In the Discussion we now write on page 23: *“Second, by combining fMRI and dMRI, and intersecting each subjects’ tracts with the GWMI directly underneath our fROIs, we were able to define the fWMT of the math and the reading networks within individual subjects. [...] In comparison, prior studies have either examined white matter6,7 or functional gray matter regions6,27 of math and reading in isolation, evaluated white and gray matter substrates in distinct groups of participants35, or have examined the white matter connections of a single region in the reading network, the VWFA18-20.”*

Generally speaking the approach is highly innovative and the findings are interesting. Additional reporting on areas where reading and addition show overlap in terms of activity would strengthen the manuscript.

Thank you for highlighting the contribution of the manuscript and for providing feedback. We believe that your suggestions have helped us strengthen the manuscript. Among other changes, we now added a conjunction analysis to **Fig. 1-5**, which includes detailed data about a region in the IOTC which shows similar level of activity during reading and math tasks.

Reviewers' Comments:

Reviewer #1:

Remarks to the Author:

The authors have addressed all of the issues I raised to the previous version of the manuscript. I particularly appreciated the inclusion of additional participants as well as the conjunction analyses on the overlap between reading and math. I think this study makes a nice contribution to the literature and I would like to congratulate the authors with this nice piece of work.

Reviewer #2:

Remarks to the Author:

After reviewing the edits and reanalyses of this manuscript, I am fully satisfied and enthusiastic about the results. I feel that the work is solid and is likely to be reproducible by other investigators. I do not have any further requests for edits or analyses from the authors.

Reviewer #3:

Remarks to the Author:

The authors have been highly responsive to the reviewers' comments and overall the manuscript has been strengthened. The other reviewers raised some important methodological aspects and it seems that they have been address, although I will defer to the other reviewers on this. The study is well motivated and the approach is novel. There are few studies in arithmetic and hence there is a need for this type of work. The results are interesting and set the stage for future studies in those with reading and/or math disabilities. At the same time I believe the authors need to be careful not to generalize from addition to arithmetic or math. Hence my comments have to do with that aspect of the manuscript.

Specifically, in the introduction the authors refer to "math" (e.g. "This approach allowed us to determine (1) what are the white matter tracts of the math and reading networks, (2) which fascicles are network unique and which contribute to both math and reading, and (3) whether white matter tracts associated with math and reading are physically intertwined or segregated within a fascicle."), but the study is limited to arithmetic, and specifically to addition. As such the language here should use the term "addition" as it cannot be assumed that the results generalize to other aspects of arithmetic (e.g. subtraction). It has been changed in other parts of the manuscript, but the same specificity should be used in the Introduction. Also the title should be reconsidered in this context.

The introduction provides a nice summary of findings in populations with disabilities, many of which are in children, so it would be important to make clear that the current study is in typical adult participants. For example " Thus, the goal of this study is twofold: (1) identify and quantify the white matter connections of cortical regions involved in 5 mathematical processing within individual subjects ..." should state "individual typical adults." Similarly it should say "data in 20 typical adult participants" instead of "data in 20 participants." Also, in the Discussion the fact that the study involved typical adult needs to be clarified. For example, it is noted "In the current study, we addressed a fundamental gap in knowledge in human brain function: what are the shared and dissociated gray and white matter substrates of math and reading, the two most essential skills every child is expected to acquire in school." It would be better to say "In the current study, we addressed a fundamental gap in knowledge in human brain function: in typical adults, what are the shared and dissociated gray and white matter substrates of math and reading, the two most essential skills every

everybody is expected to acquire.”

With that said, it would be important to know that none of the adult participants in the study had a learning disability. While it is easy to assume that they do not, one cannot be certain, especially in high IQ students. In fact, sometimes studies of this nature attract participants who harbor suspicion that they have trouble with reading or math.

Reviewer 1's comments:

The authors have addressed all of the issues I raised to the previous version of the manuscript. I particularly appreciated the inclusion of additional participants as well as the conjunction analyses on the overlap between reading and math. I think this study makes a nice contribution to the literature and I would like to congratulate the authors with this nice piece of work.

Thank you again for all your helpful suggestions that improved the paper. We are pleased that you appreciate our work.

Reviewer 2's comments:

After reviewing the edits and reanalyses of this manuscript, I am fully satisfied and enthusiastic about the results. I feel that the work is solid and is likely to be reproducible by other investigators. I do not have any further requests for edits or analyses from the authors.

Thank you for all your feedback and your enthusiasm. We feel that your feedback, which has led us to update the methodology, has significantly improved our paper.

Reviewer 3's comments:

The authors have been highly responsive to the reviewers' comments and overall the manuscript has been strengthened. The other reviewers raised some important methodological aspects and it seems that they have been address, although I will defer to the other reviewers on this. The study is well motivated and the approach is novel. There are few studies in arithmetic and hence there is a need for this type of work. The results are interesting and set the stage for future studies in those with reading and/or math disabilities. At the same time I believe the authors need to be careful not to generalize from addition to arithmetic or math. Hence my comments have to do with that aspect of the manuscript.

Specifically, in the introduction the authors refer to "math" (e.g. "This approach allowed us to determine (1) what are the white matter tracts of the math and reading networks, (2) which fascicles are network unique and which contribute to both math and reading, and (3) whether white matter tracts associated with math and reading are physically intertwined or segregated within a fascicle."), but the study is limited to arithmetic, and specifically to addition. As such the language here should use the term "addition" as it cannot be assumed that the results generalize to other aspects of arithmetic (e.g. subtraction). It has been changed in other parts of the manuscript, but the same specificity should be used in the Introduction. Also the title should be reconsidered in this context.

Thank you for your comment. We agree that our study examined addition and that further research is needed to determine in how far our findings may generalize to other forms of mathematical processing. Accordingly, we now highlight in the introduction that different arithmetic operations may vary in their neural substrates, and that we focus on addition. We now write on page 1: "*The degree to which brain activations related to math and reading overlap may also be task-dependent. For example, responses related to arithmetic fact retrieval, e.g. during adding tasks involving small numbers, such as the one used in the current study, are proposed to overlap more extensively with responses related to reading than responses induced by procedural-based computations³.*" In addition, we discuss the fact that our results may be specific to addition in the discussion, on page 16, where we write: "*Third, we focused on addition, and did not investigate neural substrates of other mathematical operations. Previous work has shown that different mathematical operations may vary in their neural substrates^{24,44,45}.*"

*Particularly, compared to other operations, addition relies more heavily on arithmetic fact retrieval⁴⁶ and thus may show more overlap with reading³. Future studies can examine which components of the revealed network are specific to addition and which components extend to other mathematical tasks.” Finally, we updated the title to highlight the fact that we studied addition, the new title is: “**Separate lanes for adding and reading in the white matter highways of the human brain.**”*

The introduction provides a nice summary of findings in populations with disabilities, many of which are in children, so it would be important to make clear that the current study is in typical adult participants. For example “ Thus, the goal of this study is twofold: (1) identify and quantify the white matter connections of cortical regions involved in 5 mathematical processing within individual subjects ...” should state “individual typical adults.” Similarly it should say “data in 20 typical adult participants” instead of “data in 20 participants.” Also, in the Discussion the fact that the study involved typical adult needs to be clarified. For example, it is noted “In the current study, we addressed a fundamental gap in knowledge in human brain function: what are the shared and dissociated gray and white matter substrates of math and reading, the two most essential skills every child is expected to acquire in school.” It would be better to say “In the current study, we addressed a fundamental gap in knowledge in human brain function: in typical adults, what are the shared and dissociated gray and white matter substrates of math and reading, the two most essential skills every everybody is expected to acquire.”

Thank you for your feedback. We implemented the three specific text edits that you recommended. In addition, we ensured that the entire manuscript consistently emphasizes the fact that we tested typically performing adults rather than children.

With that said, it would be important to know that none of the adult participants in the study had a learning disability. While it is easy to assume that they do not, one cannot be certain, especially in high IQ students. In fact, sometimes studies of this nature attract participants who harbor suspicion that they have trouble with reading or math.

Thank you for drawing our attention to this point. Behavioral performance suggests that our participants do not have learning disabilities for two reasons: (1) All participants performed both the math and reading tasks in the fMRI experiment with high accuracy. (2) We tested a subset (N=12) of our participants using the WIAT III. All of these subjects, which were recruited from our university, received almost a perfect score on both the math and reading subtests. Thus, we did not continue acquiring this form of behavioral data.

Overall, our behavioral data suggests that typical adults participated in our study, rather than adults afflicted by learning disabilities. We would be excited if future work would explore how our findings relate to dyslexia and dyscalculia as well as their high rate of co-occurrence.